# Group Robust Classification Without Any Group Information

**Christos Tsirigotis**[*]
Université de Montréal, Mila, ServiceNow Research

**Joao Monteiro**
ServiceNow Research

**Pau Rodriguez**
Apple MLR

**David Vazquez**
ServiceNow Research

**Aaron Courville**[†]
Université de Montréal, Mila, CIFAR CAI Chair

## Abstract

Empirical risk minimization (ERM) is sensitive to spurious correlations in the training data, which poses a significant risk when deploying systems trained under this paradigm in high-stake applications. While the existing literature focuses on maximizing group-balanced or worst-group accuracy, estimating these accuracies is hindered by costly bias annotations. This study contends that current bias-unsupervised approaches to group robustness continue to rely on group information to achieve optimal performance. Firstly, these methods implicitly assume that all group combinations are represented during training. To illustrate this, we introduce a systematic generalization task on the MPI3D dataset and discover that current algorithms fail to improve the ERM baseline when combinations of observed attribute values are missing. Secondly, bias labels are still crucial for effective model selection, restricting the practicality of these methods in real-world scenarios. To address these limitations, we propose a revised methodology for training and validating debiased models in an entirely bias-unsupervised manner. We achieve this by employing pretrained self-supervised models to reliably extract bias information, which enables the integration of a logit adjustment training loss with our validation criterion. Our empirical analysis on synthetic and real-world tasks provides evidence that our approach overcomes the identified challenges and consistently enhances robust accuracy, attaining performance which is competitive with or outperforms that of state-of-the-art methods, which, conversely, rely on bias labels for validation.

## 1 Introduction

Supervised learning algorithms typically rely on the empirical risk minimization (ERM) paradigm – minimizing the average loss on a training set. The ERM paradigm operates under the assumption that the training data is a representative sample of the true data distribution [67]. Consequently, models that achieve low expected loss can still be *unfair* when the model is tasked to predict outcomes for underrepresented groups [38, 10, 66], and prone to *rely on spurious correlations* between target annotations and generative attributes of the training data [32, 62] that do not hold up under more general testing conditions. As automated predictors are deployed in failure-critical applications [24] or interact with society, where fairness must be guaranteed [13, 36], robustness and fairness become requirements that standard learning strategies do not satisfy.

---

[*]Work done during internship at ServiceNow Research. Author correspondence at `tsirigoc@mila.quebec`.
[†]This work was also funded, in part, from A. Courville's Sony Research Award. Courville also acknowledges support from his Canada Research Chair.

37th Conference on Neural Information Processing Systems (NeurIPS 2023).

These limitations have motivated the search for alternatives that perform uniformly across different data subgroups [69, 61] or, equivalently, that rely less on spurious features [2] or "shortcuts" [28]. Part of solving this problem includes making the correct assumptions about data, which should consider spurious statistical correlations due to bias in its generative process. A popular example of this issue is the *cow vs. camel* classification problem [2].

As one might expect, pictures of cows very often contain a grass background, while camels are usually depicted in a desert. As such, a binary classifier that predicts whether there is grass in the background of an image could achieve a high prediction accuracy on top of natural images of the two animals. However, such a classifier would fail whenever the background changes from the typical occurrences.

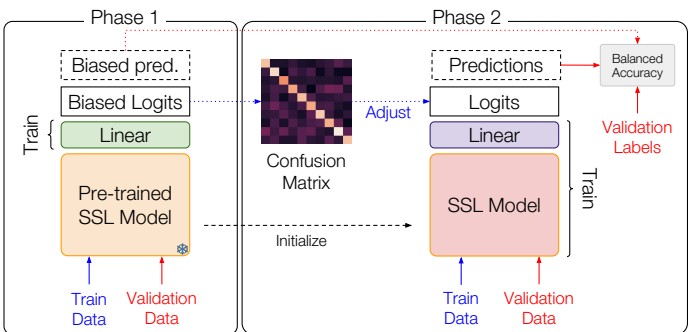

Figure 1: **Unsupervised Logit Adjustment (uLA).** We train a linear classifier on top of an SSL pre-trained model to obtain biased predictions. These predictions are then leveraged to train a debiased model. No bias information is used during training or cross-validation.

While progress has been made, existing methods to improve robustness on the least frequent attribute combinations (*e.g.*, cow on sand), require knowledge about the bias attributes (bias-supervision) either during training [6, 61], or validation [57, 48, 50]. This limits their applicability in practical scenarios where bias annotations could be too costly or impossible to obtain. Even in cases where they can be obtained somewhat efficiently, human annotators might be biased themselves or sensitive annotations may not be readily available due to privacy concerns [76].

Moreover, access to group information is rarely satisfiable on natural data, where a *curse of generative dimensionality* implies that given the high number of attributes controlling the generative process, no realistic finite sample can cover all possible - exponential in number - combinations of such attributes. In fact, many group robustness algorithms that rely on data rebalancing or loss reweighting techniques [61, 48] assume that all group combinations are present during training. However, this condition is not satisfied in systematic generalization tasks, where certain data subgroups that are present in test data, are absent from training data. This motivates us to study the extent that training algorithms can successfully operate using group information implicitly present in the training dataset.

Specifically, in the case of these challenging systematically split datasets, we study the generalization of models on unseen combinations of attributes that were independently observed during training. This combinatorial type of generalization draws its name from the cognitive property of systematicity [26]. In deep learning, this question was introduced for the first time by Lake and Baroni [46], who studied the systematicity of RNN models in sequence prediction tasks. Here, we raise the same question in classification. To do so, we introduce a benchmark consisting of systematic splits from the MPI3D (thereby named sMPI3D) image dataset [29].In particular, we use the '*real*' split of MPI3D which consists of photographs of a robotic arm that has a colored rigid object attached to its end effector. The images are captured in a way that controls their generative attributes, such as the shape of the rigid object or the position of the robotic arm. We use it to ask the following question: given a 'shape' classifier that has been trained on objects of all possible shapes and colors but only a subset of their combinations (e.g., red cubes and blue spheres), how would it perform for a new color-shape combination (e.g., blue cubes)? In this example, the 'color' attribute plays the role of a *bias*, obstructing view of all possible colors that cubes could have. In Appendix D.1 we describe in more detail the construction of the sMPI3D task, while in Figure 2d we illustrate an example of a systematic split.

As discussed above, existing state-of-the-art methods suffer from two limitations. The first one, as demonstrated by Asgari et al. [3, Table 1], is that the robust accuracy of many recent training algorithms, which do not demand any bias annotations during training, degrades severely when there is no access to bias labels during model selection. This highlights that robust algorithms should prescribe a way of performing bias-unsupervised validation, assuming the same access to i.i.d. data

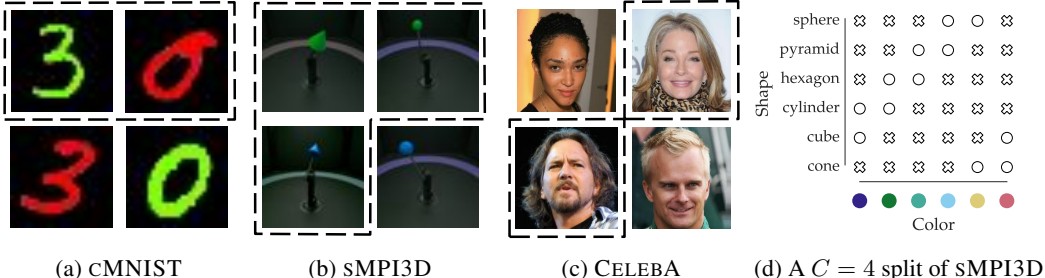

| | (a) CMNIST | (b) sMPI3D | (c) CELEBA | (d) A $C = 4$ split of sMPI3D |

Figure 2: **Left**: Example tasks. Circled in dashed lines are samples from training split, exhibiting statistically major attribute groups. Those outside are example test samples, where all groups are equally considered. A classifier trained on a biased training set may misclassify a bias-conflicting test sample - recognizing a 'red three' as 'zero' (2a) or a 'blue sphere' as a 'cone' (2b) - or be unfair to the sensitive gender attribute when tasked to classify a facial attribute, e.g. hair color (2c). **Right**: An example of a systematic split. $C$ is the number of color values per shape. 'Crosses' represent groups used to sample the training and validation splits, while 'circles' are entirely out-of-distribution.

resources as training. The second, revealed by our study on sMPI3D (Section 4.2), is that in most cases these methods fail to improve over the ERM baseline, which fails to systematically generalize.

To address these issues, our approach, summarized in Figure 1, is based on pretraining a base encoder using self-supervised learning [7] to extract a proxy for the missing bias labels and to provide an initialization point for finetuning a debiased model. Pretraining a proxy for the bias variable enables us to use this network for two purposes: first, to train a group-robust network and, second, to define a validation criterion for robust model selection. The debiasing training algorithm is based on the logit adjustment paradigm [54]. Our entirely bias-unsupervised methodology to group robustness using logit adjustment, which we call ULA, is able to compete with or outperform state-of-the-art counterparts, that otherwise utilize bias labels during model selection, in synthetic and real-world benchmarks. At the same time, it is the only method to consistently offer improvements over the ERM baseline in the sMPI3D systematic generalization task; thus effectively tackling the identified challenges about explicit or implicit use of group information.

## 2 Preliminaries

**Problem Formulation.** Consider a multi-class classification task, where $X \subset \mathcal{X}$ is the input variable and $Y \subset \mathcal{Y}$, the categorical target variable with $|\mathcal{Y}| = K$ classes. We have access to a dataset of observations $\mathcal{D} := \{(x_n, y_n)\}_{n=1}^{N}$ sampled i.i.d. from an underlying data distribution $p_{\text{data}}(X, Y)$ over $\mathcal{X} \times \mathcal{Y}$. The setting above may become problematic once we consider that the deployment data $\mathcal{D}_{\text{test}}$ are sampled from a different testing distribution: $p_{\text{test}} \neq p_{\text{data}}$. In other words, we assume that there are two data generating processes; one which generates development data ($\mathcal{D}$ and $\mathcal{D}_{\text{valid}}$, where $\mathcal{D}_{\text{valid}}$ is used for validation) according to $p_{\text{data}}$, and one which generates deployment data ($\mathcal{D}_{\text{test}}$) according to $p_{\text{test}}$.

In further detail, we focus on a particular transfer learning problem from $p_{\text{data}}$ to $p_{\text{test}}$, which is due to a distribution shift in attributes which participate in the generative process of the input variable $X$. Our study considers anti-causal prediction tasks for which the target variable $Y$ is one of the generative attributes of $X$, and $Z \subset \mathcal{Z}$ another (possibly unobserved) categorical generative attribute with $|\mathcal{Z}| = L$ classes. $Z$ is marginally independent to $Y$ under $p_{\text{test}}$, but it might not be under $p_{\text{data}}$. For this reason, we say that the variables $Y$ and $Z$ are *spuriously correlated* in training and validation data, and $Z$ will also be referred to as the *bias attribute*. Under this setting, a group $(y, z) \in [K] \times [L]$ is defined as a combination of target and bias attribute values, and group robustness can be formulated as a transfer learning problem from $p_{\text{data}}$ to $p_{\text{test}}$ under the following two assumptions relating the two joint distributions over $X, Y$ and $Z$:

$$p_{\text{test}}(y, z) \propto \mathbb{1}_{\text{Supp } p_{\text{data}}(Y) \times \text{Supp } p_{\text{data}}(Z)}(y, z), \tag{1}$$

$$\text{and } p_{\text{test}}(x|y, z) = p_{\text{data}}(x|y, z), \tag{2}$$

where $\mathbb{1}_S(s) = 1$ if $s \in S$ else 0, the characteristic function of a set $S$, and $\text{Supp } p$ denotes the support set of a distribution $p$. Relation 1 asserts that, during test time, the target and bias variables

are distributed uniformly over the product of their respective marginal supports under $p_{\text{data}}$. In other words, all combinations of observed attributes values are considered equally. This also implies that $Y$ and $Z$ are marginally independent under $p_{\text{test}}$[3]. On the other hand, relation 2 assumes the invariance of mechanism, which is an assumption typically found in *disentangled causal process* literature [64]. These two assumptions underlie the group robustness literature, in that they are equivalent to the evaluation of classifiers under a popular robust performance criterion which is described below.

**Performance Criteria.** Let $\hat{y}(x; f) \coloneqq \operatorname{argmax}_y f(x)_y$ be the predictions of a scoring function $f : \mathcal{X} \to \mathbb{R}^K$. The accuracy of $f$ under $p_{\text{test}}$ corresponds to the *group-balanced accuracy*

$$\operatorname{Acc}_{Y|X}(f; p_{\text{test}}) = \underset{\substack{y,z \sim p_{\text{test}} \\ x \sim p_{\text{test}}(\cdot|y,z)}}{\mathbb{E}} \mathbb{1}\big(y = \hat{y}(x; f)\big) = \frac{1}{KL} \sum_{y,z} \underset{x \sim p_{\text{data}}(\cdot|y,z)}{\mathbb{E}} \mathbb{1}\big(y = \hat{y}(x; f)\big), \quad (3)$$

which is a frequently used performance metric in literature [61, 48, 50]. Essentially, this performance criterion first separately computes the accuracy for samples $x$ of each group $(y, z)$ and then averages them. The averaging operation per individual group accuracy directly stems from the uniformity assumption in $p_{\text{test}}$ (rel. 1) and implements a group fairness notion: we care equally about performing well in all groups. Another popular group robustness criterion is the *worst-group accuracy* [61] which substitutes the average accuracy over individual groups with the minimum (worst) accuracy.

As described in Section 5, there exist different approaches to improve group fairness [61, 48, 50] that depend on knowing biases for cross-validation. Here we focus on logit adjustment techniques like the one proposed by Liu et al. [50], which co-train two models where one corrects the biases of the other. In the following sections, we show how, by decoupling the training of the biased model from the bias-corrected model, it is possible to obtain a proxy criterion that can be used for cross-validation without knowing the bias attribute.

**Logit Adjustment** was originally developed as a technique for supervised learning under class-imbalanced [25, 53] or long-tailed [54] data. For clarity of presentation, we will assume for the moment that we have access to bias labels. We describe sLA, a bias-supervised training process with logit adjustment. Let $h_\theta : \mathcal{X} \to \mathbb{R}^{|\mathcal{Y}|}$ be the model that we want to train for group robustness, such as a neural network parameterized by $\theta$ implementing a function from samples in $\mathcal{X}$ to the unnormalized logit space of $Y$. We then train by minimizing the average cross-entropy loss for the following *logit adjusted model*

$$p_\theta(y|x, z) \propto \exp(h_\theta(x)_y + \log \hat{p}_{\text{data}}(y|z)). \tag{4}$$

We estimate the conditional $\hat{p}_{\text{data}}(y|z)$ directly from the available training data, for example via empirical frequencies of the finite number of groups $(y, z) \in [K] \times [L]$. Finally, during inference, we cancel out the contribution of the logit bias term $\hat{p}_{\text{data}}(y|z)$ and predict only according to the optimized neural network $h_{\theta*}$.

Note that, since $Y$ is a categorical variable, $\log \hat{p}_{\text{data}}(y|z)$ takes non-positive values. For this reason, we can intuitively interpret logit adjustment as a soft masking operation for outputs of $h_\theta$ which are unlikely in the training data when we have observed $z$. In this way, we account for the dependency of $Y$ to $Z$ which spuriously exists in the training distribution. By fitting the cross-entropy objective under logit adjustment, the network $h_\theta$ has to model the remaining relations for $Y \mid X$ that are not spurious since those are already accounted for. An expressive enough model class in Equation (4) can achieve this, assuming further that the likelihood ratio $\frac{p_{\text{data}}(x|y,z)}{p_{\text{data}}(x|z)} = \frac{p_{\text{data}}(y|x,z)}{p_{\text{data}}(y|z)}$ is independent of $z$. In appendix A, we derive sLA from first principles, bridging the gap between the well-studied application of this technique to the class-imbalance problem with its application to group robustness. Moreover, we demonstrate that sLA is a well-justified procedure by proving the following proposition.

**Proposition 2.1** (sLA optimizes the group-balanced accuracy)**.** *Under the assumption that the hypothesis class $p_\theta(y|x, z)$ (eq. 4) contains $p_{\text{data}}(y|x, z)$, the minimizer network $h_{\theta*}$ of the cross-entropy loss maximizes a lower bound to the group-balanced accuracy $\operatorname{Acc}_{Y|X}(h_{\theta*}; p_{\text{test}})$.*

**Systematic Generalization.** We expect sLA models to systematically generalize, as they optimize for the transfer learning problem defined by Equations (1) and (2). As Figure 2d suggests, systematic generalization is just an extreme case of such problems, for which samples from some combinations

---

[3]Generally, the variables, which are uniform over a support which factorizes into a cartesian product of their marginal supports, are independent: $c\mathbb{1}_{\mathcal{X} \times \mathcal{Y}}(x, y) = c\mathbb{1}(x, y \in \mathcal{X} \times \mathcal{Y}) = c\mathbb{1}(x \in \mathcal{X})\mathbb{1}(y \in \mathcal{Y})$.

have $p_{\text{data}}(y, z) = 0$ during training, however they populate the test set with $p_{\text{test}}(y, z) > 0$. As a counterexample, the setting by Ahmed et al. [1] does not correspond to a systematic generalization task, as we define it. In what they refer to as systematic splits of colored MNIST, all possible combinations of color and digit are exposed to the model during its training, while in our sMPI3D some color and shape combinations are only revealed during test-time. This detail makes our setting significantly more challenging, as the conditions for applying importance sampling are not met, as $\text{Supp} \, p_{\text{data}}(Y, Z) \not\supseteq \text{Supp} \, p_{\text{test}}(Y, Z)$ [4, Chapter 5]. This means that a simple re-weighting of the per-sample loss with $\frac{1}{p_{\text{data}}(y,z)}$, in order to estimate $\mathbb{E}_{p_{\text{test}}}[l_{\text{ce}}(y, h_\theta(x))]$, is not appropriate for the task.

**Self-Supervised Learning (SSL)** refers to a collection of methods [7] in unsupervised representation learning where unlabeled data provides the supervision by defining tasks where a model is asked to predict some part of the input from other parts, for example by contrasting independently transformed versions of data samples [14, 35, 15]. Methods like SimCLR [14], MoCo [35], BYOL [30] and Barlow Twins [73] provide solutions for low-data generalization, robustness, as well as transferability of learnt representations for image classification. Although self-supervised learning methods are generally more robust to distribution shifts than purely supervised methods [49], they can still be affected by significant shifts, for example in long-tailed data learning [5, 63]. In this work, we explore the limits and utility of SSL in group robust classification. An extended preliminary discussion about self-supervised algorithms can be found in Appendix B.

# 3 ULA: Bias-unsupervised Logit Adjustment

Here, we introduce bias-unsupervised logit adjustment (ULA), a logit correction approach which improves on the work of Liu et al. [50] by *removing dependency on explicit bias annotations both during training and validation*. For this reason, we create a proxy variable for the bias via the predictions of a pretrained network using SSL. The advantage of pretraining, over co-training a bias network [57, 17, 50], is two-fold: First, we can reuse the fixed bias proxy network to define a validation criterion for group robustness. This enables us to perform hyperparameter search, but also informs us about when to stop training the debiased model, which is critical for optimal performance [48]. Second, recent literature [39, 43] has demonstrated, in the case of training with bias-supervised data, that a linear classifier on top of pretrained base models provides with substantial group-robustness improvements. Here, we initialize the debiased model from the pretrained base model and finetune it using logit adjustment. Figure 1 provides a summary of our approach and its pseudocode can be found in Appendix C. Further training details are discussed in the following.

## 3.1 Bias-unsupervised Training

**Biased network: pretrain with SSL.** We start by training a *base model*, $f_{\text{base}}$, using an SSL method on the unlabeled data of the training set $\mathcal{D}$. We decide on the hyperparameters of the SSL algorithm (learning rate, weight decay, temperature, augmentations and potentially others) by maximizing the i.i.d. validation accuracy of an online linear classifier, which probes the representations of the base model. Afterwards, we train a linear classifier $g_\phi$ on top of a frozen $f_{\text{base}}$ and against target variable labels $Y$ using a vanilla cross-entropy loss, in order to derive a proxy for the bias variable. Finally, we retrieve the composite neural network $h_{\text{bias}} = g_\phi \circ f_{\text{base}}$, and use its predictions $\hat{y}(x; h_{\text{bias}})$ as a proxy for the missing bias variable observations. For the same purpose, Nam et al. [57] trained a bias-extracting network by employing Generalized Cross Entropy [75], an objective which provides robustness to noisy labels. Since the goal of the proxy network is to predict the spurious attribute, bias-conflicting or minority samples can be perceived as mislabeled data points. Here, we follow these observations and leverage the representations learnt with SSL which, when composed with a low-capacity (linear) classifier, provide with a model that is more robust to label-noise [71]. In addition to deriving a bias proxy, we will also use $f_{\text{base}}$ as an initialization point in the parameter space for training the debiased model.

In this paper, we have chosen to use MoCoV2+ [15] as the SSL algorithm for the image classification tasks we consider. We make this choice since contrastive learning algorithms, like the MoCo family [35, 15], offer relatively stable training since they explicitly prevent representation collapse in their loss function (see Appendix B). However, as we demonstrate in Section 4.3, our method is not restrained to the use of a particular SSL algorithm.

**Debiased network: logit adjustment.** In the absence of bias labels during training, we need a substitute for the estimate $\hat{p}_{\text{data}}(Y, Z)$ in eq. 4. We use the predictions $y_{\text{bias}}$ of the bias proxy network $h_{\text{bias}}$ to that end. The resulting joint distribution between the target variable and the biased network's predictions, $\hat{p}_{\text{data}}(y, y_{\text{bias}})$, can be thought of as a soft confusion matrix of $h_{\text{bias}}$ and can be computed using the available training data with

$$\hat{p}_{\text{data}}(y, y_{\text{bias}}) = \frac{1}{|\mathcal{D}|} \sum_{x', y' \in \mathcal{D}} p_{\text{bias}}(y_{\text{bias}} \,|\, x') \mathbb{1}(y = y'), \tag{5}$$

where $p_{\text{bias}}(y_{\text{bias}} \,|\, x) \propto \exp(h_{\text{bias}}(x)_{y_{\text{bias}}}/\tau)$ is the biased model conditional. Note that $p_{\text{bias}}(y_{\text{bias}} \,|\, x)$ is post-hoc calibrated by a temperature hyperparameter $\tau$. As we rely on the biased network to approximate the spurious correlation structure described in $p_{\text{data}}(Y, Z)$, it is crucial that the predicted conditional probabilities of the biased network are calibrated correctly [31, 56, 53]. Afterwards, we are ready to begin training a debiased network with logit adjustment as in Section 2; using only the predictions of $h_{\text{bias}}$ as the bias variable. The debiased network is initialized at the composition of a random linear classifier with the pretrained network $f_{\text{base}}$, and during training we finetune it while adjusting its output logits by

$$p_\theta(y \,|\, x) \propto \exp\left( h_\theta(x)_y + \eta \log \hat{p}_{\text{data}}\big(y \,|\, \hat{y}(x; h_{\text{bias}})\big)\right), \tag{6}$$

where $\eta \geq 0$ is a hyperparameter controlling the strength of the additive logit bias. Notice that for $\eta = 0$ we fall back to ERM training. By tuning $\eta$ we can mitigate calibration errors of the debiased model [53], similar to what $\tau$ does for the bias proxy. Selected hyperparameter configurations can be found in Appendix E.

## 3.2 Bias-unsupervised Validation

We re-purpose the pretrained biased classifier $h_{\text{bias}}$ so that training no longer requires bias-annotated validation data for model selection. Our bias-unsupervised validation criterion calculates a balanced accuracy across pairs $(y, y_{\text{bias}}) \in [K] \times [K]$ of true labels and biased classifier predictions. In practice, we compute

$$\widehat{\text{BalAcc}}(f; h_{\text{bias}}) := \frac{1}{K^2} \sum_{y, y_{\text{bias}}} \frac{1}{|S_{y, y_{\text{bias}}}|} \sum_{x_i, y_i \in S_{y, y_{\text{bias}}}} \mathbb{1}\big(y_i = \hat{y}(x_i; f)\big) \tag{7}$$

$$S_{y, y_{\text{bias}}} := \left\{ (x_i, y_i) \in \mathcal{D}_{\text{valid}} \,|\, y_i = y \text{ and } \hat{y}(x_i; h_{\text{bias}}) = y_{\text{bias}} \right\},$$

where $S_{y, y_{\text{bias}}}$ are partitions of $\mathcal{D}_{\text{valid}}$ based on the value of predictions of $h_{\text{bias}}$ on a sample $x_i$ and its ground-truth target label $y_i$. This corresponds to a form of group-balanced accuracy. Alternatively, we could also calculate a form of worst-group accuracy by taking the minimum across $S_{y, y_{\text{bias}}}$. We find that worst-case validation is more suitable for tasks with small number of classes $K$.

During training, we evaluate models at every epoch and we select the one that maximizes our validation score across the duration of a training trial. In addition, we use this criterion to tune hyperparameters. In particular, for each task we tune learning rate, weight decay, logit adjustment strength coefficient $\eta$, calibration temperature $\tau$ and, in addition, the number of pretraining steps for the SSL backbone - whenever it is applicable - and for the linear classification probe of the bias proxy network. Fig. 1 depicts our bias-unsupervised training and validation procedures.

## 4 Experiments

**Datasets.** The tasks we consider are all specific instances of the setup above (see Fig. 2 and Section 2). This spans group robustness challenges like colored MNIST [57, CMNIST], corrupted CIFAR10 [37, CCIFAR10] and WATERBIRDS [62], fair classification benchmarks like CELEBA [51], and systematic generalization tasks such as the contributed sMPI3D. Details about their construction can be found in Appendix D.

**Training Setup.** For CMNIST, we train a 3-hidden layer MLP, while we use a ResNet18 [34] for CCIFAR10 and sMPI3D, and a ResNet50 for WATERBIRDS and CELEBA. For all datasets except WATERBIRDS, we pretrain the base model with the MOCOV2+ [15] process, while training of the linear probe for the bias network and finetuning for the logit adjusted debiased network happen with

| | Bias Labels | | cMNIST | | | | cCIFAR10 | | | |
|---|---|---|---|---|---|---|---|---|---|---|
| | Train | Val | 0.5% | 1.0% | 2.0% | 5.0% | 0.5% | 1.0% | 2.0% | 5.0% |
| [61] GROUPDRO[†] | ✓ | ✓ | 63.12 | 68.78 | 76.30 | 84.20 | 33.44 | 38.30 | 45.81 | 57.32 |
| [57] LFF* | ✗ | ✓ | $52.50_{\pm2.43}$ | $61.89_{\pm4.97}$ | $71.03_{\pm2.44}$ | $80.57_{\pm3.84}$ | $28.57_{\pm1.30}$ | $33.07_{\pm0.77}$ | $39.91_{\pm0.30}$ | $50.27_{\pm1.56}$ |
| [17] DFA* | ✗ | ✓ | $65.22_{\pm4.41}$ | $81.73_{\pm2.34}$ | $84.79_{\pm0.95}$ | $89.66_{\pm1.09}$ | $29.95_{\pm0.71}$ | $36.49_{\pm1.79}$ | $41.78_{\pm2.29}$ | $51.13_{\pm1.28}$ |
| [50] LC[†] | ✗ | ✓ | $71.25_{\pm3.17}$ | $\mathbf{82.25_{\pm2.11}}$ | $\mathbf{86.21_{\pm1.02}}$ | $91.16_{\pm0.97}$ | $\mathbf{34.56_{\pm0.69}}$ | $37.34_{\pm0.69}$ | $47.81_{\pm2.00}$ | $54.55_{\pm1.26}$ |
| ERM* | ✗ | ✗ | $35.19_{\pm3.49}$ | $52.09_{\pm2.88}$ | $65.86_{\pm3.59}$ | $82.17_{\pm0.74}$ | $23.08_{\pm1.25}$ | $25.82_{\pm0.33}$ | $30.06_{\pm0.71}$ | $39.42_{\pm0.64}$ |
| uLA (ours) | ✗ | ✗ | $\mathbf{75.13_{\pm0.78}}$ | $81.80_{\pm1.41}$ | $84.79_{\pm1.10}$ | $\mathbf{92.79_{\pm0.85}}$ | $34.39_{\pm1.14}$ | $\mathbf{62.49_{\pm0.74}}$ | $\mathbf{63.88_{\pm1.07}}$ | $\mathbf{74.49_{\pm0.58}}$ |

Table 1: Results using datasets from Chu et al. [17] for various % of bias-conflicting examples in the training set. We report avg. group-balanced test accuracy (%) and std. dev. over 5 seeds. *Results from Chu et al. [17]. [†]Results from Liu et al. [50].

| | Bias Labels | | WATERBIRDS | | CELEBA | |
|---|---|---|---|---|---|---|
| | Train | Val | i.i.d. | worst group | i.i.d. | worst group |
| [61] GROUPDRO* | ✓ | ✓ | 93.5 | 91.4 | 92.9 | 88.9 |
| [9] ERM | ✗ | ✓ | 97.6 | 86.7 | 93.1 | 77.8 |
| [57] LFF* | ✗ | ✓ | 97.5 | 75.2 | 86.0 | 77.2 |
| [48] JTT* | ✗ | ✓ | 93.6 | 86.0 | 88.0 | 81.1 |
| [50] LC | ✗ | ✓ | - | $90.5_{\pm1.1}$ | - | $88.1_{\pm0.8}$ |
| ERM | ✗ | ✗ | 97.3 | 72.6 | 95.6 | 47.2 |
| Bardenhagen et al. [9] | ✗ | ✗ | 97.5 | 78.5 | 88.0 | 78.9 |
| [3] MASKTUNE | ✗ | ✗ | $93.0_{\pm0.7}$ | $86.4_{\pm1.9}$ | $91.3_{\pm0.1}$ | $78.0_{\pm1.2}$ |
| uLA (ours) | ✗ | ✗ | $91.5_{\pm0.7}$ | $86.1_{\pm1.5}$ | $93.9_{\pm0.2}$ | $86.5_{\pm3.7}$ |

Table 2: Results on WATERBIRDS and CELEBA. We report avg. test accuracy (%) and std. dev. over 5 seeds. *Results from Liu et al. [48].

AdamW [52] optimizer. For WATERBIRDS instead, we leverage a base model which was pretrained on Imagenet [60], following baselines in the literature for fair comparison, and we finetune it using SGD. Finally, for cMNIST, cCIFAR10 and sMPI3D we use our group-balanced bias-unsupervised validation criterion, whereas for WATERBIRDS and CELEBA the worst-group version. Further details are described in Appendix E.

**Baselines.** We compare uLA with vanilla ERM and a diverse set of group robustness techniques described in Section 5. GROUPDRO [61] provides with a fully bias-supervised baseline, while LFF [57], JTT [48], LC [50] and DFA [17] are bias-unsupervised during training although they require bias annotations during validation to achieve robust optimal performance. We also consider two fully bias-unsupervised methods: Bardenhagen et al. [9] propose early stopping networks to derive a proxy for bias-unsupervised validation, and MASKTUNE [3] which provide competitive results under fully bias-unsupervised benchmarks without performing any validation procedure.

### 4.1 Results on Benchmarks

In Table 1, we report the group-balanced accuracy on cMNIST and cCIFAR10, across different percentages of bias-conflicting examples in the training set. For cMNIST, we observe that our method performs overall competitively against LFF, DFA and LC, even though these baselines use bias annotations during model selection. On the other hand, for cCIFAR10, we observe a significantly improved group robust performance for 3/4 difficulty levels. The highest difference is observed at the 1.0% task, where our method outperforms GROUPDRO by about 24% absolute increase in group-balance accuracy.

In Table 2, we observe worst-group accuracy results in the more challenging WATERBIRDS and CELEBA datasets. In both cases, our approach performs again competitively among bias-unsupervised training algorithms that leverage bias information during validation, falling slightly behind LC, which is the best out of the ones considered. Notably, our approach is still the best performing fully bias-unsupervised method, outperforming the supervised learning pretraining validation scheme of Bardenhagen et al. [9], and performing on par with MASKTUNE on WATERBIRDS and better than it on CELEBA by $\approx$8% absolute worst-group accuracy.

| | Bias Labels | sMPI3D (various $C$ colors / shape) | | | |
|---|---|---|---|---|---|
| | Train & Val | 2 | 3 | 4 | 5 |
| % i.i.d. samples | - | 33.33 | 50.00 | 66.66 | 83.33 |
| [61] GROUPDRO | ✓ | $31.23_{\pm1.88}$ | $46.01_{\pm4.13}$ | $69.68_{\pm7.82}$ | $82.18_{\pm8.39}$ |
| ERM | ✗ | $31.94_{\pm1.67}$ | $47.68_{\pm4.06}$ | $71.94_{\pm7.71}$ | $83.10_{\pm8.64}$ |
| [48] JTT | ✗ | $31.89_{\pm0.88}$ | $48.93_{\pm2.04}$ | $67.78_{\pm3.14}$ | $83.31_{\pm3.51}$ |
| [50] LC* | ✗ | $31.29_{\pm0.96}$ | $45.01_{\pm2.31}$ | $61.67_{\pm5.06}$ | $94.62_{\pm0.88}$ |
| ULA (ours) | ✗ | $\mathbf{59.58}_{\pm3.12}$ | $\mathbf{80.53}_{\pm3.16}$ | $\mathbf{91.11}_{\pm3.52}$ | $\mathbf{98.05}_{\pm0.64}$ |

Table 3: Results on sMPI3D for various numbers of $C$ colors per shape value in the training set (see Appendix D). We report avg. group-balanced accuracy (%) and std. dev. over 5 seeds & dataset generations. *GroupMix augmentation was not used for fairness of comparison.

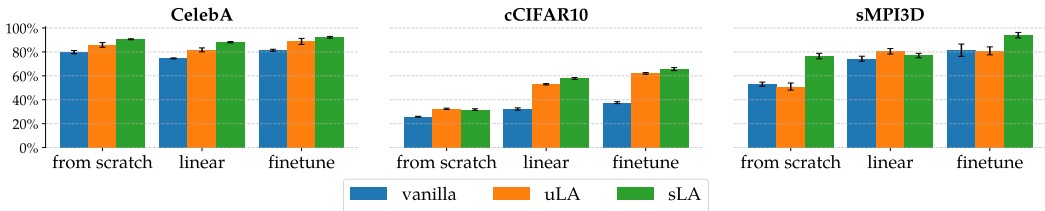

Figure 3: Ablations on the influence of SSL pretraining and on the use of VANILLA process, SLA, or ULA for training against the downstream task. For cCIFAR10 (1%) and sMPI3D ($C = 3$) we report the group-balanced test accuracy, whereas for CELEBA the worst-group. We report avg. accuracies and std. dev. over 5 seeds.

## 4.2 Systematic Generalization

sMPI3D is our contributed task which we use to study combinatorial systematicity in classifiers. With this benchmark, we aim to study the ability of classifiers to generalize to samples generated from novel combinations of observed generative attributes values, which under $p_{\text{data}}$ have $0$ probability. The target task is to classify the *shape* of an object, which is spuriously correlated with its *color*. We devise 4 difficulty levels for this task which we denote by $C$, the number of color values present in the training set per shape. In Fig. 2d, we display a possible split between in-distribution and o.o.d. combinations of attributes for $C = 4$. Details about its construction are given in Appendix D.

**Results.** In Table 3, we further validate our approach on our contributed systematic generalization task. Under this setting, all bias-unsupervised approaches are evaluated fairly since their models are validated with exactly the same data resources; access to bias labels cannot give model selection advantage to any algorithm since there are no o.o.d. samples in the validation (just like in the training) split. Our method is the only one which consistently offers group-balanced accuracy improvements across difficulty levels, demonstrating generalization to o.o.d. samples. On the other hand, GROUPDRO, vanilla ERM, JTT and LC are not able to increase group-balanced accuracy over the percentage of i.i.d. samples present in the balanced test set.

## 4.3 Ablation Studies

We perform a set of studies to understand better the efficacy of our approach. In Fig. 3, we ablate the choice of training paradigm and the influence of a pretrained base model using SSL to the robust performance of a trained model to the target task. The paradigms we choose are among vanilla cross-entropy minimization, ULA and SLA (bias-supervised logit adjustment - see Section 2). For this ablation, a bias-supervised validation procedure was used for comparison against the fully bias-supervised SLA baseline. We find that finetuning the pretrained base model gives the best performance across training paradigms and tasks. Second, forn CELEBA and cCIFAR10, vanilla cross-entropy finetuning does not offer stark performance improvement over vanilla training from scratch. Only when we apply a logit adjustment training procedure, we are able to take significant advantage of the learnt representation space. At the same time, the accuracy gaps between ULA and

|  | Group-balanced Test Acc. |
|---|---|
| ERM | $25.82_{\pm 0.33}$ |
| GROUPDRO | $38.30$ |
| ULA W. MOCOV2+ | $62.49_{\pm 0.74}$ |
| ULA W. BYOL | $59.73_{\pm 2.03}$ |
| ULA W. BARLOWTWINS | $50.08_{\pm 1.23}$ |

Table 4: Ablation of SSL pretraining methods for the backbone model. Default $\eta = 1.0$ and $\tau = 1.0$ are used. We pretrain for 1000 epochs, and train the linear head of the bias proxy for 100 epochs. We report avg. group-balanced test accuracy (%) and std. dev. over 5 seeds on CCIFAR10 (1%).

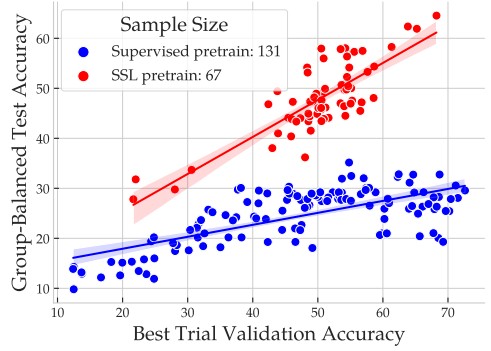

Figure 4: Validation vs test on CCIFAR10 (1%).

| Pretraining method | Pearson correlation |
|---|---|
| End-to-end supervised | $0.690$ $(0.588, 0.770)$ |
| SSL + linear head | $0.819$ $(0.721, 0.885)$ |

Table 5: Effect of pretraining method for a bias proxy network on results of hyperparameter search using our validation criterion. Each training trial's hyperparameters were sampled from the same prior. Pretraining with SSL results in trials whose best validation criterion correlates better with selected model's group-balanced test accuracy on CCIFAR10 (1%). In parenthesis, we compute 95% confidence intervals.

SLA are small which indicates that we are able to recover correctly the bias attribute with the proxy network. On the other hand, for SMPI3D the gaps between ULA and SLA are large. This shows that, in the systematic generalization case, bias extraction remains an open challenge for future work, and that improvement over baseline procedures in Table 3 is due to pretraining with SSL.

In addition, we study the impact of the choice of SSL method that we use to pretrain the backbone. For this reason, we perform an ablation experiment on CCIFAR10 (1%) by changing the pretraining strategy from MOCOV2+ to BYOL [30] or BARLOW TWINS [73]. We find that, while BYOL performs within the error margin of the best MOCOV2+ setting, BARLOW TWINS underperforms. BARLOW TWINS seeks to match the empirical (i.i.d.) cross-correlation between features to the identity. Arguably, we expect that the cross-correlation is different under the shifted test set. In any case, ULA significantly outperforms the non-ULA baselines with any of the considered SSL methods.

Finally, we study how the choice of pretraining paradigm for the bias network influences the quality of hyperparameter search using our proposed validation criterion. In Fig. 4, we present two separate searches on the same space of hyperparameters using two different pretraining approaches. In red we see our approach of pretraining an SSL base model for the bias network, and in blue we see a baseline approach where we pretrain with purely supervised learning. We observe that SSL pretraining enables stronger correlation between the proposed bias-unsupervised group-balanced validation criterion and the corresponding test accuracy on CCIFAR10. It is more difficult to tune hyperparameters with a bias network pretrained with supervised learning, because it may fit the training set entirely. In that case, the validation criterion collapses to the in-distribution test accuracy which is not indicative of the group-balanced test accuracy. On the contrary, classification with a linear probe on top of SSL representations prevents from fitting the training set entirely, having small generalization gaps in-distribution [8]. In this way, the validation criterion remains strongly correlated even in larger validation accuracy values, maintaining its utility in a greater range of hyperparameter configurations.

## 5 Related Work

Prior literature can be grouped according to three main strategies that attempt to improve a model's robustness to dataset bias [54]. (i) *Resampling* strategies increase or decrease the frequency of biased

attributes in the input space [27, 55] or latent space [23, 74, 17]. (ii) *Loss reweighting* methods balance class or feature importances during training [74, 21, 12, 65, 40]. Especially relevant to our work are the reweighting methods that improve robustness and generalization when training on biased datasets [61, 57, 17, 48]. (iii) *Post-hoc adaptation* methods [54, 25, 19, 42, 41, 3, 70] correct the biases learned by already-trained models. Most relevant to our work is the logit adjustment technique proposed by Menon et al. [54] for long-tail learning, which leverages class frequencies to rebalance the model predictions after training or to train with a loss function which is aware of the class prior. We develop a logit adjustment technique for the problem of learning from biased data, which does not require previous knowledge about the dataset's biases.

Several solutions have emerged for the group robustness problem, especially when bias attribute data is available. For example, GROUPDRO [61] leverages explicit bias attributes to reduce worst-case group loss. Our method, however, reduces reliance on such data, recognizing that access to them can be impractical. Bias-unsupervised methods like Learning from Failure (LFF) [57] and Disentangled Feature Augmentation (DFA) [17] reweight the loss of the unbiased model using a co-trained biased model, removing the need for bias supervision. Conversely, Just-train-twice (JTT) [48] reweights misclassified samples from an initial biased training, emphasizing worst-performing group data. Similarly, He et al. [33] exemplify the same intuition in that a second network is trained on examples that cannot be predicted already using the spurious structure. Liu et al. [50] proposed a bias-unsupervised logit adjustment technique (LC) also based on co-training a biased network. Utilizing domain knowledge, Clark et al. [18] describe a bias-supervised logit adjustment approach to debiasing Visual Question Answering (VQA) models by incorporating a bias proxy which is trained exclusively on question data. These methods only generalize to seen attribute groups during training and require bias knowledge during validation for optimal performance. We tackle these issues by employing SSL to pre-train a network, deriving a bias proxy for debiased model training and validation, as well as using it as initialization.

In order to derive bias-unsupervised solutions, literature has proposed to train reference models as proxies for the missing bias labels. Creager et al. [20] optimize a reference model for group assignments which maximally violate a relaxation of the Environment Invariance Criterion. Chen et al. [16] seek to establish conditional independence between the predictions of the proxy model and the target variable given the inferred groups. Our work follows more closely the approach of [48], in which a biased network is simply trained with ERM. As we demonstrate at Figure 6 of Appendix F, by utilizing a frozen backbone pretrained with SSL, our approach improves on the sensitivity to the number of training steps for the bias proxy.

Bardenhagen et al. [9] suggested a validation scheme dependent on early stopping of bias proxy network training. Our method, using SSL pretraining, avoids this by treating pretraining steps as tunable hyperparameters, maintaining performance of alternatives which were otherwise tuned with bias information. Chen et al. [16] perform experiments using a methodology dubbed as Training Environments Validation (TEV). Similar to us, TEV validates models based on inferred groups from training, however the methodology is unfortunately not well documented in the literature, making its reproducibility difficult. As we show in the ablation study of Figure 4, implementation details can make a large difference in the quality of the criterion. Finally, MASKTUNE [3] eliminates spurious shortcuts by masking input data during a secondary training phase. Despite its resilience in performance without using a bias-unsupervised o.o.d. model selection criterion, the need for a reliable validation strategy for group robustness remains.

## 6  Conclusion

We explored group robust classification in synthetic and real tasks, proposing a generalization task with unseen attribute combinations. Current robust classification methods struggle in this setting, motivating our SSL-based logit adjustment approach. Importantly, we introduce **a methodology for training and validating robust models without group labels**. Empirical evaluations on five datasets show our method outperforms existing fully bias-unsupervised approaches and rivals those using bias annotations during validation. In terms of *limitations and broader impact* of our contributions, as machine learning systems handle high-stakes applications, ensuring robustness to underrepresented samples is crucial. Our work reduces reliance on known data biases, but existing benchmarks differ from real-life scenarios with unknown biased attribute combinations. To bridge this gap, we proposed a synthetic benchmark and encourage further research on real data, revealing more system limitations.

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

# Appendix

## A    Bias-Supervised Training with Logit Adjustment

The goal of this Appendix section is to develop a bias-supervised approach for maximizing the group-balanced accuracy (Equation (3)), which is going to serve as an initial point in the development of our entirely bias-unsupervised methodology in Section 3.

Under the problem formulation we introduced in Section 2, the metric that we would ultimately like to maximize given a scoring function $f_{Y|X} : \mathcal{X} \to \mathbb{R}^K$ is $\mathrm{Acc}_{Y|X}(f_{Y|X}; p_{\text{test}})$, the top-1 accuracy under the test distribution $p_{\text{test}}$. Simple algebraic manipulations reveal that this test accuracy corresponds to the group-balanced accuracy:

$$\mathrm{Acc}_{Y|X}(f_{Y|X}; p_{\text{test}}) := \mathop{\mathbb{E}}_{x,y \sim p_{\text{test}}} \mathbb{1}\big(y = \mathop{\mathrm{argmax}}_{y'} f_{Y|X}(x)_{y'}\big) \tag{8}$$

$$= \mathop{\mathbb{E}}_{\substack{y,z \sim p_{\text{test}} \\ x \sim p_{\text{test}}(\cdot|y,z)}} \mathbb{1}\big(y = \mathop{\mathrm{argmax}}_{y'} f_{Y|X}(x)_{y'}\big)$$

$$= \frac{1}{KL} \sum_{(y,z) \in [K] \times [L]} \mathop{\mathbb{E}}_{x \sim p_{\text{data}}(\cdot|y,z)} \mathbb{1}\big(y = \mathop{\mathrm{argmax}}_{y'} f_{Y|X}(x)_{y'}\big).$$

A related quantity in the literature is the class-balanced top-1 accuracy, which is used in class-imbalanced classification problems [11, 53]. We can make the connection by imagining $(y, z)$ as a multi-label classification target in a multi-label, but class-imbalanced, problem. The multi-label class-balanced accuracy of a multi-label scoring function $f_{Y,Z|X} : \mathcal{X} \to \mathbb{R}^K \times \mathbb{R}^L$ would then be

$$\mathrm{Acc}_{Y,Z|X}(f_{Y,Z|X}; p_{\text{test}}) = \frac{1}{KL} \sum_{(y,z) \in [K] \times [L]} \mathop{\mathbb{E}}_{x \sim p_{\text{data}}(\cdot|y,z)} \mathbb{1}\big((y,z) = \mathop{\mathrm{argmax}}_{y',z'} f_{Y,Z|X}(x)_{y',z'}\big). \tag{9}$$

We will set towards building a multi-label classifier which maximizes multi-label class-balanced accuracy and we will finally show that its part which only predicts $Y$ from $X$ can be shown to maximize a lower bound to the group-balanced accuracy.

We assume that $\mathcal{D}$ and $\mathcal{D}_{\text{valid}}$ provide us with tuples of observations $(x, y, z)$ distributed according to $p_{\text{data}}$. In this case, results in the class-imbalance literature [53, 44, 19] state that the Bayes-optimal scoring function $f^*_{Y,Z|X}$ for maximizing the class-balanced criterion (9) is given by

$$\mathop{\mathrm{argmax}}_{y,z} f^*_{Y,Z|X}(x)_{y,z} = \mathop{\mathrm{argmax}}_{y,z} p_{\text{test}}(x \,|\, y, z). \tag{10}$$

Note that under the invariance of mechanism assumption (2) we have that $p_{\text{test}}(x \,|\, y, z) = p_{\text{data}}(x \,|\, y, z)$. This allows us to compute the mechanism conditional probability in $p_{\text{data}}$ distribution terms

$$p_{\text{test}}(x \,|\, y, z) = \frac{p_{\text{data}}(y, z \,|\, x)}{p_{\text{data}}(y, z)} p_{\text{data}}(x) \tag{11}$$

by utilizing the Bayes rule. This way we have a candidate strategy to maximize multi-label class-balanced accuracy in Equation (9): First, we will estimate $p_{\text{data}}(y, z \,|\, x)$ and $p_{\text{data}}(y, z)$ from data, and then we will divide the estimates. Notice that, in order to make a prediction about $(y, z)$ given $x$, we do not need to model $p_{\text{data}}(x)$ as this quantity is constant in $(y, z)$, and thus does not influence the argmax.

**Training process.** We can directly estimate $p_{\text{data}}(y, z)$ from available training data by computing the empirical frequencies of pairs $(y, z) \in [L] \times [K]$, or by a Bayesian estimate with a Dirichlet prior. On the other hand, we can train a parameterized discriminative model $p_\theta$ with maximum conditional likelihood estimation to estimate $p_{\text{data}}(y, z|x)$

$$\max_\theta \frac{1}{|\mathcal{D}|} \sum_{x,y,z \in \mathcal{D}} \log p_\theta(y, z \,|\, x) \tag{12}$$

which is equivalent to the minimization of a KL divergence estimate between the $p_{\text{data}}$ and $p_\theta$ model conditionals.

The method described so far would have sufficed if the goal was to perform multi-label classification, however we need to devise a way to overcome the bias-supervision, since we only care about classifying $Y$ without having access to $Z$ annotations. For this reason, we need to proceed to modelling assumptions about $p_\theta$, which will allow us to develop the bias-unsupervised method described in Section 3. First, we recall that $p_{\text{data}}$ factorizes as

$$p_{\text{data}}(y, z \mid x) = p_{\text{data}}(y \mid x, z) p_{\text{data}}(z \mid x). \tag{13}$$

Let $h_\theta : \mathcal{X} \to \mathbb{R}^K$ be a parameterized model (such as a neural network) tasked to predict unnormalized logits of $Y$ given an observation $x$. Then, we model $p_{\text{data}}(y \mid x, z)$ by

$$p_\theta(y \mid x, z) \propto \exp\left(h_\theta(x)_y + \log \hat{p}_{\text{data}}(y \mid z)\right). \tag{14}$$

Unpacking our modelling assumption Equation (14), we have the following: First, we are going to compute $\hat{p}_{\text{data}}(y \mid z) = \frac{\hat{p}_{\text{data}}(y,z)}{\hat{p}_{\text{data}}(z)}$, by using our estimated $\hat{p}_{\text{data}}(y, z)$ and by marginalizing out $y$ for $\hat{p}_{\text{data}}(z) = \sum_{y \in [K]} \hat{p}_{\text{data}}(y, z)$. Given a tuple of observables $(x, y, z)$ in a training batch, we use $\log \hat{p}_{\text{data}}(y \mid z)$ to adjust additively the outputs of our parametric discriminative model $h_\theta$ to the observed input $x$. Notice that since $Y$ is a categorical variable, $\log \hat{p}_{\text{data}}(y \mid z) \in (-\infty, 0]$ for all $y \in [K]$. For this reason, we can intuitively interpret *logit adjustment* as a soft masking operation for outputs of $h_\theta$ which are unlikely in the training data when we have observed $z$. In this way, we account for the dependency of $Y$ to $Z$ which spuriously exists in the training distribution. By fitting the cross-entropy objective under logit adjustment, the network $h_\theta$ has to model the remaining relations for $Y \mid X$ that are not spurious since those are already accounted for.

**Prediction process.** Our modelling assumption enables the prediction of $y$ without computing the maximizer over combinations of $(y, z) \in [K] \times [L]$. This is an important step towards developping a bias-unsupervised training procedure. This is because assumption 14 allows us to compute the maximizer for $y \mid x$ and $z \mid x$ separately. We show that under our modelling assumption, if we are only interested in predicting only the target variable $y$ given an observation $x$, we do not need to model $z \mid x$ at all. To understand how this is possible, we take a close look at the total expression for optimal predictor. After fitting the model $p_\theta$ using Equation (12), we calculate the estimated scoring function for the multi-label class-imbalanced problem as in Equation (11)

$$\hat{f}_{Y,Z \mid X}(x) = \hat{p}_{\text{test}}(x \mid y, z) = \frac{\hat{p}_{\text{data}}(y, z \mid x)}{\hat{p}_{\text{data}}(y, z)} \hat{p}_{\text{data}}(x) \tag{15}$$

$$= \frac{p_\theta(y \mid x, z)}{\hat{p}_{\text{data}}(y \mid z)} \frac{\hat{p}_{\text{data}}(z \mid x)}{\hat{p}_{\text{data}}(z)} \hat{p}_{\text{data}}(x)$$

$$= \frac{\exp\left(h_\theta(x)_y + \cancel{\log \hat{p}_{\text{data}}(y \mid z)}\right)}{Z_\theta(z, x) \cancel{\hat{p}_{\text{data}}(y \mid z)}} \frac{\hat{p}_{\text{data}}(z \mid x)}{\hat{p}_{\text{data}}(z)} \hat{p}_{\text{data}}(x)$$

$$= \exp\left(h_\theta(x)_y\right) \frac{\hat{p}_{\text{data}}(z \mid x) \hat{p}_{\text{data}}(x)}{Z_\theta(z, x) \hat{p}_{\text{data}}(z)},$$

where $Z_\theta(z, x) := \sum_{y \in [K]} \exp\left(h_\theta(x)_y + \log \hat{p}_{\text{data}}(y \mid z)\right)$ is the partition function of $p_\theta(y \mid x, z)$. Notice that the expression simplifies into a multiplication of two terms; the first, $\exp\left(h_\theta(x)_y\right)$, solely depends on $(x, y)$, while the second fraction only depends on $(x, z)$. Then, predicting according to Equation (10) amounts to computing

$$\max_{y, z} \hat{p}_{\text{test}}(x \mid y, z) = \max_y \exp\left(h_\theta(x)_y\right) \max_z \frac{\hat{p}_{\text{data}}(z \mid x) \hat{p}_{\text{data}}(x)}{Z_\theta(z, x) \hat{p}_{\text{data}}(z)}. \tag{16}$$

Essentially, information about $Z$ is only need during training time and not during prediction of $Y$ given $X$. In practice, during prediction we adapt the model by removing the logit adjustment term in order to acquire the unbiased network for the task. Consequently, the obtained scoring function for debiased $Y$ predictions is just the trained neural network

$$\hat{f}_{Y \mid X}(x; \theta) := h_\theta(x). \tag{17}$$

Notice that our design decisions are well-justified, in the sense that, under our class of models (eq. 14), we have $\text{Acc}_{Y,Z \mid X}(\hat{f}_{Y,Z \mid X}; p_{\text{test}}) \leq \text{Acc}_{Y \mid X}(\hat{f}_{Y \mid X}; p_{\text{test}})$. This means that by maximizing the multi-label class-balanced accuracy we indirectly maximize the top-1 accuracy of the candidate scoring function $\hat{f}_{Y \mid X}$ under the test distribution $p_{\text{test}}$, which is the group-balanced accuracy as we have demonstrated.

**Proposition A.1** (sLA optimizes the group-balanced accuracy). *Under the assumption that the hypothesis class of $p_\theta(y|x, z)$ (eq. 4) contains $p_{data}(y|x, z)$, the minimizer network $h_{\theta^*}$ of the cross-entropy loss maximizes a lower bound to the group-balanced accuracy $\mathrm{Acc}_{Y|X}(h_{\theta^*}; p_{test})$.*

*Proof.* We will start by the quantity

$$\mathrm{Acc}_{Y,Z|X}\big(f; p_{\text{test}}(\cdot|y,z)\big) := \mathbb{P}^{(\text{test})}_{x|y,z}\big((y,z) = \operatorname*{argmax}_{y',z'} f(x)_{y',z'}\big) \tag{18}$$

the classification accuracy conditioned on the pair $(y, z)$. Recall that under our modelling assumption Equation (14), the estimated Bayes-optimal scoring function for the multi-label class-imbalanced classification problem is

$$\hat{f}_{Y,Z|X}(x) = \hat{p}_{\text{test}}(x \mid y, z) = \exp\big(h_{\theta^*}(x)_y\big)C(z,x), \tag{19}$$

where $C(z,x) := \frac{\hat{p}_{\text{data}}(z|x)\hat{p}_{\text{data}}(x)}{Z_\theta(z,x)\hat{p}_{\text{data}}(z)}$ is a constant in $y$. We then substitute to get

$$\mathrm{Acc}_{Y,Z|X}\big(\hat{f}_{Y,Z|X}; p_{\text{test}}(\cdot|y,z)\big) = \mathbb{P}^{(\text{test})}_{x|y,z}\Big((y,z) = \operatorname*{argmax}_{y',z'} \exp\big(h_{\theta^*}(x)_{y'}\big)C(z',x)\Big). \tag{20}$$

However $\max_{z'}$ gets pushed to the innermost term since $\exp\big(h_{\theta^*}(x)_{y'}\big)$ is independent of $z'$, as well as $\max_{z'} C(z',x)$ is independent of $y'$. This means that we can take the argmax of individual expression independently

$$\mathrm{Acc}_{Y,Z|X}\big(\hat{f}_{Y,Z|X}; p_{\text{test}}(\cdot|y,z)\big) = \tag{21}$$
$$\mathbb{P}^{(\text{test})}_{x|y,z}\big(\{y = \operatorname*{argmax}_{y'} \exp\big(h_{\theta^*}(x)_{y'}\big)\} \cap \{z = \operatorname*{argmax}_{z'} C_2(z',x)\}\big),$$

and since $\mathbb{P}(A \cap B) \le \mathbb{P}(A)$ and $\exp$ is a strictly increasing function, we get

$$\mathrm{Acc}_{Y,Z|X}\big(\hat{f}_{Y,Z|X}; p_{\text{test}}(\cdot|y,z)\big) \le \mathbb{P}^{(\text{test})}_{x|y,z}\big(y = \operatorname*{argmax}_{y'} h_{\theta^*}(x)_{y'}\big). \tag{22}$$

We recognise the scoring function $\hat{f}_{Y|X}(x) := h_{\theta^*}(x)$ on the right hand side of the expression. Finally, by summing up all inequalities for all $(y, z) \in [K] \times [L]$, and dividing by $KL$, we get

$$\frac{1}{KL}\sum_{y,z}\mathrm{Acc}_{Y,Z|X}\big(\hat{f}_{Y,Z|X}; p_{\text{test}}(\cdot|y,z)\big) \le \frac{1}{KL}\sum_{y,z}\mathbb{P}^{(\text{test})}_{x|y,z}\big(y = \operatorname*{argmax}_{y'} \hat{f}_{Y|X}(x)_{y'}\big) \tag{23}$$

$$\mathrm{Acc}_{Y,Z|X}(\hat{f}_{Y,Z|X}; p_{\text{test}}) \le \mathrm{Acc}_{Y|X}(\hat{f}_{Y|X}; p_{\text{test}}). \tag{24}$$

This shows that the scoring function $\hat{f}_{Y|X}(x) := h_{\theta^*}(x)$ maximizes a lower bound to the group-balanced accuracy. $\qquad\square$

## B  Self-supervised Learning Algorithms

| | Target Attribute | | Bias Attribute | |
|---|---|---|---|---|
| | $p_{\text{data}}$ | $p_{\text{test}}$ | $p_{\text{data}}$ | $p_{\text{test}}$ |
| cMNIST (1%) | 98.86 | 27.01 | 99.88 | 96.04 |
| cCIFAR10 (1%) | 98.93 | 36.53 | 99.60 | 82.05 |
| sMPI3D ($C = 3$) | 98.19 | 66.08 | 99.91 | 87.22 |
| CELEBA$^*$ | 94.01 | 75.44 | 95.74 | 90.60 |

Table 6: Downstream online linear classification against the *target* and *bias* attributes of each task considered (see Appendix D). We present top-1 (average) accuracy (%) of the corresponding classifiers evaluated at last epoch of MoCoV2+ training, under samples from the i.i.d. validation set ($p_{\text{data}}$) and the o.o.d. test set ($p_{\text{test}}$). $^*$For CELEBA, since the test split is not controlled to be group-balanced, we simulate the accuracy under $p_{\text{test}}$ by utilizing the existing bias annotations to estimate a group-balanced test accuracy.

In our work, we are using a contrastive learning algorithm [58, 14, 35, 15] to pretrain a base model for the bias proxy, as well as to initialize the debiased model. In contrastive learning, an instance discrimination pretext task is defined from unlabeled data. In particular, pairs of data points (positives), which are derived by independently augmenting the same observation (views), should have representations that are closer in distance than representations of other samples in the dataset (negatives). To achieve that we optimize a form of the InfoNCE loss [58]

$$\mathcal{L}(q, k^+, \{k_i^-\}_{i=1}^N) = -\log \frac{\exp(q^\top k^+/\tau)}{\exp(q^\top k^+/\tau) + \sum_{i=1}^N \exp(q^\top k_i^-/\tau)}, \tag{25}$$

where $(q, k^+)$ is the positive pair and $\{k_i^-\}_{i=1}^N$ are the negatives. Typically, representations are projected to the unit hypersphere for training stability. In the MoCoV2+ framework [15], gradients are backpropagated only through the query representation ($q$) to the encoding network, while the key representations ($k^+$ and $k^-$) are extracted by a network derived from the exponential moving average of its parameters. Finally, positive key representations of past batches are kept in a queue in memory to serve as negative key representations for the subsequent batches.

## C   uLA: Algorithm

In Algorithm 1, we provide a high-level description of the ULA methodology for training and validation of group-robust models without any bias annotations. In addition, a PyTorch [59] implementation is available at the following repository: `https://github.com/tsirif/uLA`.

## D   Datasets

### D.1   Systematic MPI3D

sMPI3D is our contributed dataset which we use to study combinatorial systematicity in classifiers. By this term, we designate the ability of classifiers to generalize to samples generated from novel combinations of generative attributes, which under $p_{\text{data}}$ have 0 probability. However, all the constituent values of the individual generative attributes in the novel combinations have been observed under some combination in the training data. The *real* split of MPI3D [29] consists of photographs of a robotic arm that has a colored rigid object attached to its end effector. The images are captured in a way that controls their generative attributes, such as the shape of the rigid object or the position of the robotic arm.

We use this dataset to create development, and deployment splits to test for systematic generalization. In particular, we consider the shape of the rigid object to be the target variable $Y_{\text{shape}}$, while its color to be the bias $Z_{\text{color}}$. There are 6 possible shapes for the objects and 6 possible colors, totaling 36 combinations of attributes. Figure 2d illustrates pairs of attributes used to sample an example systematic training split. A training set is created by assigning $C$ number of colors per shape so that: first, $p_{\text{data}}(z_{\text{color}} \mid y_{\text{shape}}) = \frac{1}{C}$ if $(y_{\text{shape}}, z_{\text{color}})$ exists in the elected pairs of attributes for training; otherwise 0. Second, we make sure that marginally all colors and shapes are represented uniformly, which means that $p_{\text{data}}(y) = \frac{1}{6}$ and $p_{\text{data}}(z) = \frac{1}{6}$. During testing, we desire to generalize to samples generated from $p_{\text{test}}$ which distributes pairs $(y_{\text{shape}}, z_{\text{color}})$ uniformly, covering combinations that are entirely missing from the training set. In our benchmarks, we are testing methods for $C \in \{2, 3, 4, 5\}$, against the same 5 sets of systematic splits which we generate independently per $C$.

**Systematic split generation.** As we described above, sMPI3D is generated using the *real* "split" of the MPI3D dataset [29]. The dataset is distributed as a tensor of shape $(6, 6, 2, 3, 3, 40, 40, 64, 64, 3)$, with the last 3 axis corresponding to pixel values of a $64 \times 64$ RGB image and the rest to coordinates for generative attribute values. The first two coordinates correspond to the color and shape attributes for the rigid object depicted in the image. The procedure for generating splits of sMPI3D is stochastic, meaning that it depends on a random seed set by the user. The reason for this is that we would like to benchmark methods against the possibility that some systematic splits are consistently easier to generalize than others. To generate a train-validation-test split of sMPI3D, we first generate the pattern of included-excluded combinations for the development dataset (training and validation datasets), like the one presented in Figure 2d. The pattern depends on the argument $C$, the number of color values observed per shape value. Then, we randomly permute data points across the the first two

**Algorithm 1:** ULA: Logit adjustment without bias labels during training or model selection.

**Data:** Training split: $\mathcal{D} := \{x_m, y_m\}_{m=1}^{M}$

**Data:** Validation split: $\mathcal{D}_{\text{valid}} := \{x_n, y_n\}_{n=1}^{N}$

**Data:** Hyperparameters: SSL checkpoint $T_{\text{ssl}}$, linear probe training steps $T_{\text{stop}}$, logit adjustment strength $\eta$ and calibration $\tau$, and optimization hyperparameters

**Result:** Debiased model $h_{\theta^*}$ and its `validation_score`

**1**
```
/* Load pretrained SSL encoder                                          */
```
**2** $f_{\text{base}} \leftarrow \text{load}(T_{\text{ssl}})$
**3**
```
/* Train Linear Head                                                    */
```
**4** Define a classifier $h_{\text{bias}} = g_\phi \circ f_{\text{bias}}$ from the composition of pretrained $f_{\text{base}}$ and linear classifier $g_\phi$ with parameters $\phi$
**5** Predict biased predictions using $\hat{y}(x; h_{\text{bias}}) = \text{argmax}_y\, h_{\text{bias}}(x)_y$
**6** **for** $t = 1$ **to** $T_{stop}$ **do**
**7** $\quad$ Sample batch $B$ from $\mathcal{D}$
**8** $\quad$ Update $\phi$ so that linear model on top of $h_{\text{bias}}(x)$ minimizes average cross-entropy on $B$
**9** **end**
**10**
```
/* Finetune with logit adjustment                                       */
```
**11** Define calibrated model of biased predictions $p_{\text{bias}}(y_{\text{bias}}|x) \propto \exp\left(h_{\text{bias}}(x)_{y_{\text{bias}}}/\tau\right)$
**12** Compute confusion matrix $\hat{p}_{\text{data}}(y, y_{\text{bias}})$ `// See Equation (5)`
**13** Define $h_\theta$ to be the debiased model
**14** Initialize base model in $h_\theta$ from pretrained $f_{\text{base}}$
**15** Predict debiased predictions using $\hat{y}(x; h_\theta) = \text{argmax}_y\, h_\theta(x)_y$
**16**
**17** `checkpoints` $\leftarrow [\,]$
**18** **for** $t = 1$ **to** *maximum number of iterations* **do**
**19** $\quad$ Sample batch $B$ from $\mathcal{D}$
**20** $\quad$ Update $\theta$ so that model $p_\theta(y|x) \propto \exp\left(h_\theta(x)_y + \eta \log \hat{p}_{\text{data}}\left(y|\hat{y}(x; h_{\text{bias}})\right)\right)$ minimizes average cross-entropy on $B$
**21** $\quad$ `validation_score` $\leftarrow$ `compute_validation_criterion`$(\mathcal{D}_{\text{valid}}, h_{\text{bias}}, h_\theta)$ `// see` Equation (7)
**22** $\quad$ `checkpoints` $\leftarrow$ `checkpoints` $+ [(\text{validation\_score}, h_\theta)]$
**23** **end**
**24** Final debiased model, $h_{\theta^*} \leftarrow \max_{\text{validation\_score}}$ `checkpoints`

---

axis of the data tensor, which correspond to the color and shape attributes. Afterwards, we split the data tensor according to whether a certain position $(\text{color}, \text{shape})$ belongs to the development dataset or not. We subsample the deployment dataset uniformly across i.i.d. combinations for $180k$ and $18k$ mutually exclusive data points for the training and validation sets respectively, for all $C \in \{2, 3, 4, 5\}$. Finally, the remaining data points are combined and $54k$ images are sampled uniformly from all combinations to create the unbiased test set. We provide code for generating systematic splits on MPI3D in the supplementary material.

### D.2 Non-systematic Benchmarks

**Colored MNIST.** (CMNIST) is an RGB version of MNIST dataset [47], in which the digit target variable $Y_{\text{digit}}$ is paired with a color bias variable $Z_{\text{color}}$ to draw an image of that digit using a specific color. In this work, we study the dataset which follows the data generation procedure from Nam et al. [57]. Each of the digits $k \in \{0, ..., 9\}$ is paired with a distinct color out of a choice of ten, with probability $p_{\text{data}}(Z_{\text{color}} = k \,|\, Y_{\text{digit}} = k) = 1 - \beta$ of the available training samples for that digit $k$. The rest of the probability mass is split uniformly across the remaining color options, $p_{\text{data}}(Z_{\text{color}} = l \,|\, Y_{\text{digit}} = k) = \frac{\beta}{9}$ for $l \neq k$. Essentially, $\beta$ controls for the percentage of bias-conflicting samples in the training set. The goal is to train a classifier that performs well under the

|  | Learning Rate | Batch Size | Weight Decay | Temperature |
|---|---|---|---|---|
| CMNIST | 1.0 | 256 | 1e-3 | 0.1 |
| CCIFAR10 | 0.9 | 256 | 1e-4 | 0.1 |
| SMPI3D | 0.9 | 256 | 1e-4 | 0.1 |
| CELEBA | 0.3 | 128 | 3e-5 | 0.1 |
| SSL Augmentations | | | | |
| CMNIST | RRC + Gaussian Blur | | | |
| CCIFAR10 | RRC + Color Jitter + Gray Scale + Gaussian Blur + Horizontal Flip | | | |
| SMPI3D | RRC + Color Jitter + Gray Scale + Gaussian Blur + Horizontal Flip | | | |
| CELEBA | RRC + Color Jitter + Solarization + Horizontal Flip | | | |

Table 7: Hyperparameters used during SSL pretraining.

test set in which digits are paired with colors with random chance $1/10^2$. We obtain four different tasks by varying $\beta \in \{0.5\%, 1\%, 2\%, 5\%\}$.

**Corrupted CIFAR10.** (CCIFAR10) is a modification of the CIFAR10 dataset [45] by Hendrycks and Dietterich [37], in which images are affected by a type of texture noise which correlates with the original target category. Similarly to CMNIST, there are 10 different types of texture noise each of which is predominantly paired to a unique CIFAR10 label. The training set is created under $p_{\text{data}}(Z_{\text{noise}} = k \,|\, Y_{\text{cifar}} = k) = 1 - \beta$ with the rest of probability spread uniformly to the bias-conflicting options of texture noise. Our goal again is to perform well under the test set in which pairs of texture noise and labels are distributed uniformly. Four development datasets are used in the experiments with $\beta \in \{0.5\%, 1\%, 2\%, 5\%\}$.

**Waterbirds.** Our study employs the WATERBIRDS dataset, devised by Sagawa et al. [61]. This dataset is derived from the CUB dataset's bird images [68], superimposed on backgrounds from the Places dataset [77]. The dataset classifies seabirds and waterfowl as waterbirds, and all other species as landbirds. Ocean and natural lake backgrounds from Places are labeled as 'water background', while bamboo and broadleaf forest backgrounds are termed 'land background'. Four groups emerge: land background with waterbird and landbird, and water background with waterbird and landbird. The first two are minority groups due to fewer examples, while the latter two constitute majority groups. The dataset maintains the original training, validation, and test splits from Sagawa et al. [61]. In training, 95% of waterbirds and landbirds are respectively paired with water and land backgrounds, while we care for the worst-group test accuracy.

**CelebA.** We examine a fair classification task using the CELEBA dataset [51] of celebrity facial attributes. Same to Sagawa et al. [61], we consider the *BlondHair* attribute to be the target variable. Then, a spurious correlation can naturally be observed with the binarized gender attribute *Male*. Four groups emerge for which the minorities are males with blond hair, followed by females with non-blond hair. We use the standard splits defined in the literature, and we report the worst-group test accuracy.

## E  Experiment Details

**Architecture, Optimization, Augmentation and Validation.** For CMNIST, we train a 3-hidden layer MLP with 100 hidden neurons for each layer, while we use a ResNet18 [34] for CCIFAR10 and SMPI3D, and a ResNet50 for WATERBIRDS and CELEBA. For all datasets except WATERBIRDS, we pretrain the base model with the MOCOV2+ [15] process, while for WATERBIRDS we use an already pretrained ResNet50 on Imagenet [60]. Training of the linear probe for the bias network and finetuning for the logit adjusted debiased network happen with AdamW [52] optimizer using default momentum hyperparameters $\beta = (0.9, 0.999)$, except for WATERBIRDS where we use SGD with $0.9$ momentum. The learning rate for all dataset, excluding WATERBIRDS, is scheduled with a cosine decaying rule. Batch size is set to 256 for CMNIST, CCIFAR10, SMPI3D and CELEBA, and 64 for WATERBIRDS. We use minimal data augmentation for training the linear probe, as well as the debiased logit adjusted model, following the baselines we compare against for fair comparison. In more details, we use random resized crops (RRC) and random horizontal flipping for all datasets, except CMNIST where no augmentation is used. Finally, training happens for a maximum of 100

|  | Learning Rate | Weight Decay | $\eta$ |
|---|---|---|---|
| CMNIST | $\{5\text{e-}4, 1\text{e-}3, 2\text{e-}3, 5\text{e-}3\}$ | $\{0.0, 1\text{e-}4, 1\text{e-}3, 1\text{e-}2, 1\text{e-}1\}$ | $\{1.0, 1.25, 1.5\}$ |
| CCIFAR10 | $\{1\text{e-}5, 5\text{e-}5, 1\text{e-}4, 5\text{e-}4, 1\text{e-}3\}$ | $\{0.0, 1\text{e-}4, 1\text{e-}3, 1\text{e-}2, 1\text{e-}1\}$ | $\{1.0, 1.25, 1.5, 1.75, 2.0\}$ |
| SMPI3D | $\{1\text{e-}5, 5\text{e-}5, 1\text{e-}4, 5\text{e-}4, 1\text{e-}3\}$ | $\{0.0, 1\text{e-}4, 1\text{e-}3, 1\text{e-}2, 1\text{e-}1\}$ | $1.0$ |
| WATERBIRDS | $\{1\text{e-}4, 5\text{e-}4, 1\text{e-}3, 5\text{e-}3\}$ | $\{0.0, 1\text{e-}4, 1\text{e-}2\}$ | $\{1.0, 1.5, 2.0, 2.5, 3.0\}$ |
| CELEBA | $\{1\text{e-}5, 5\text{e-}5, 1\text{e-}4, 5\text{e-}4\}$ | $\{0.0, 1\text{e-}5, 1\text{e-}4, 1\text{e-}3, 1\text{e-}2\}$ | $\{1.0, 1.25, 1.5\}$ |

|  | $\tau$ | $T_{\text{ssl}}/T_{\text{stop}}$ |
|---|---|---|
| CMNIST | $\{0.1, 1.0, 2.0\}$ | $\{10, 20, 50, 100\}$ epochs |
| CCIFAR10 | $\{0.5, 1.0, 2.0\}$ | $\{100, 200, 300, 400, 500\}$ epochs |
| SMPI3D | $1.0$ | $\{10, 20, 50, 100\}$ epochs |
| WATERBIRDS | $\{0.1, 0.5, 1.0, 2.0\}$ | $\{50, 100, 200, 500, 1000, 2000, 5000\}$ steps |
| CELEBA | $\{0.5, 1.0, 2.0\}$ | $\{10, 20, 50, 100\}$ epochs |

Table 8: Search space used to tune hyperparameters for each of the datasets considered

epochs for all datasets (except WATERBIRDS which we train for 200), and the best model across the training duration is selected for each trial, by evaluating our proposed validation score at the end of each epoch and monitoring for the maximum value. For CMNIST, CCIFAR10 and SMPI3D, we use our bias-unsupervised group-balanced validation score (see Equation (7)), whereas for WATERBIRDS and CELEBA the corresponding worst-group version.

**Self-supervised Pretraining.** For all cases, we adapt self-supervised pretraining recipes from *solo-learn* software library [22]. In particular, we optimize the MOCOV2+ loss with a LARS optimizer [72] using $\eta_{\text{LARS}} = 0.002$. Furthermore, an MLP projector with a single hidden layer is used to project features to a hypersphere in a 256-dimensional space. Cosine scheduling is used for the learning rate and for the exponential moving average coefficient of the momentum encoder. We train for a maximum of 100 epochs for the CMNIST, SMPI3D and CELEBA tasks, and for 500 epochs for the CCIFAR10 datasets. We save periodically checkpoints of base models across training, as we are going to decide which one to use as a basis for the bias proxy and for the debiased model as a hyperparameter of ULA. For each dataset, we perform a small random search over contrastive temperature and data augmentation by maximizing the i.i.d. validation accuracy of an online linear classifier on-top of extracted representations. In general, we fix the contrastive temperature to $0.1$ and use default Imagenet augmentations for SMPI3D and CCIFAR10, while we search for effective variants of those for CMNIST and CELEBA. Details about the hyperparameter combinations used are displayed at Table 7.

In all cases, we verify that the augmentations at Table 7 **do not cause representations to be invariant to variations of the bias attribute**. We take extra care in order to delineate that, especially for synthetic tasks, improvements are observed due to our ULA methodology, and not due to the use of data augmentations which nullify the bias attribute; something which, in real-life tasks such as CELEBA, might be impossible to do with hand-crafted transformation. We classify against the bias attribute using a linear classifier on extracted representations, and we find that, for the hyperparameters used, bias information is indeed retained (see Table 6).

**Hyperparameter Search for ULA.** For each dataset and task, we search over the respective spaces defined at Table 8. In particular, we sample uniformly 64 independent configurations (128 for WATERBIRDS) and we choose the one with the best bias-unsupervised validation score (Equation (7)). To make the space a bit smaller, we consider the same weight decay for the training of the bias proxy and the debiased logit adjusted network, as well as we consider the same number of training epochs

|  | Learning Rate | Weight Decay | $\eta$ | $\tau$ | $T_{\text{ssl}}$ (epochs) | $T_{\text{stop}}$ |
|---|---|---|---|---|---|---|
| CMNIST (1%) | 2e-3 | 1e-3 | 1.5 | 2.0 | 100 | 100 epochs |
| CCIFAR10 (1%) | 5e-5 | 1e-4 | 1.0 | 1.0 | 500 | 500 epochs |
| SMPI3D ($C = 3$) | 5e-5 | 1e-4 | 1.0 | 1.0 | 100 | 100 epochs |
| WATERBIRDS | 1e-3 | 0.0 | 3.0 | 0.1 | - | 50 steps |
| CELEBA | 1e-4 | 0.0 | 1.5 | 0.5 | 10 | 10 epochs |

Table 9: Best hyperparameters selected by our proposed bias-unsupervised validation score.

for SSL-pretrained base network and the linear head for the bias proxy (which we train online in our implementation). For WATERBIRDS, the linear head is trained on top of a pretrained ResNet50 on Imagenet, which is taken from a PyTorch [59] model repository. The pretrained model's parameters remain frozen during the training of the bias proxy, so in this case there is no number of pretraining steps to select. The hyperparameters used for the benchmarking experiments, that we presented in Section 4, can be found at Table 9.

## F  Extended Experiments

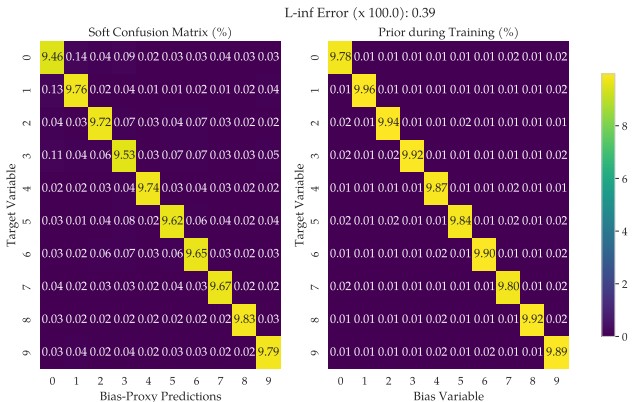
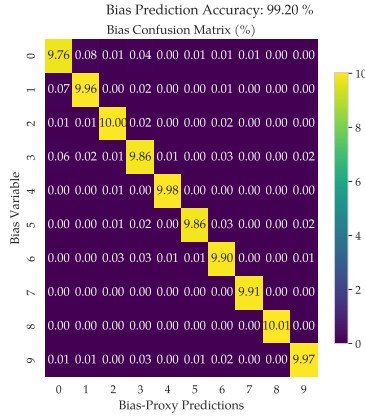

(a) **Left:** Soft confusion matrix estimate using $p_{\text{bias}}(y_{\text{bias}}|x)$ of the bias proxy and the formula of Eq. 5 in the main paper. **Right:** Estimated prior distribution between the target and bias variables using ground-truth bias observations.

(b) Accuracy of predicting the bias variable via $\arg\max_y p_{\text{bias}}(y_{\text{bias}}|x)$, even though the bias proxy was trained on target variables.

Figure 5: Heatmaps displaying the effectiveness of the proposed method to derive a proxy network for predicting the bias variable. Backbone is pretrained using BYOL on CCIFAR10 (1%).

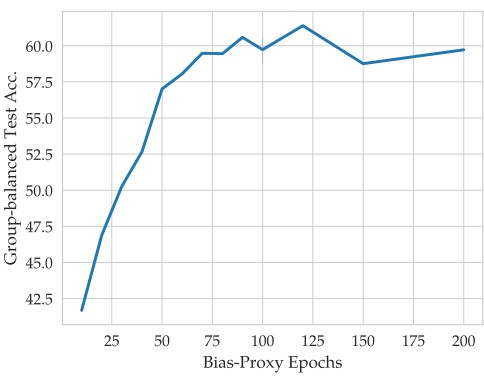
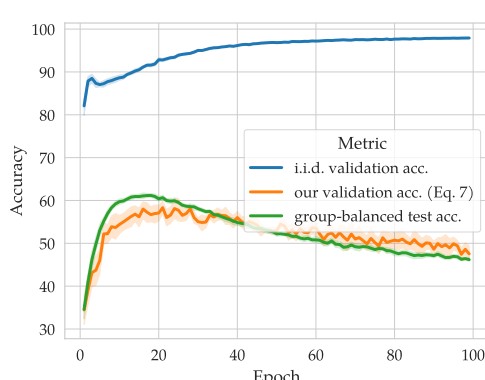

Figure 6: **Left:** Ablation of final group-balanced test accuracy (%) on CCIFAR10 (1%) with respect to the number of epochs we train the linear head on-top of a pretrained backbone with BYOL to produce the proxy network for the bias variable. **Right:** Progression of accuracy metrics (%) during training of a debiased network with ULA. Our validation accuracy (orange) tracks better the group-balanced test accuracy (green), than the standard in-distribution validation accuracy. Model selection with our criterion will result in a better model (close to the mode of the test curve) compared to the one using i.i.d. validation.

