# OpenReview forum: "Group Robust Classification Without Any Group Information"
_NeurIPS.cc/2023/Conference — NeurIPS 2023 poster_

### Official Review · Reviewer_Tfdv · 2023-06-26

**Soundness:** 3 good
**Presentation:** 3 good
**Contribution:** 4 excellent
**Rating:** 6
**Confidence:** 4

**Summary:**

The paper identifies that current methods tackling spurious correlations requires group annotation in either the training or validation stage. To address this limitation, the authors propose uLA, a bias-unsupervised method that achieves superior empirical performance without any group annotation.


**Strengths:**

The paper studies an important problem: how to tackle spurious correlations when group annotation is unavailable. The proposed algorithm is simple yet effective, demonstrating superior empirical improvement. The performance on the systematic generalization is particularly promising as it is a much more realistic scenario.


**Weaknesses:**

Although the authors show some conceptual and empirical advantage of SSL over pure supervised training, it remains unclear how crucial a role SSL has in the training pipeline. I hope the authors can provide clearer insights about the criterion of selecting the pretraining scheme for the shared backbone model. (Also see question 3)

The robustness of the algorithm to different hyperparameter settings is questionable (also see question 2). Not only does the biased prediction matter, but the biased logits are also quite important in the training pipeline. However, the logits can take dramatically different values given different hyperparameters. I hope the authors can provide more insights in this regard and perhaps include more principled method for hyperparameter search.


**Questions:**

1. For the reported baseline, how are the models trained? Do they also go through the MOCOV2+ pretraining and linear probing as uLA?
2. On line 225, it is mentioned that the model needs to be validated at every epoch, which is uncommon in practice. Could you provide some justification for this procedure? Does it mean that the algorithm is sensitive to hyperparameters?
3. Why use an Imagenet pretrained model on waterbird? Are there any scenarios where using SSL hurt the performance? I hope the authors can provide more detail about this design choice and more discussion about the selection of the training scheme of the shared backbone.


**Limitations:**

The authors have adequately addressed the limitation.

---

> ### Author Rebuttal · Authors · 2023-08-10
>
> We thank the reviewer for their detailed assessment of our work and for highlighting the merits of our approach, as well as the importance of the problem. We address all concerns below:
>
> ### Weakness 1/Question 3.b.: Even though SSL is demonstrated to provide benefits over pure supervised learning, it is not clear how the choice of SSL pretraining affects debiased training.
>
> In **Table 1 of the rebuttal material**, we demonstrate results of uLA on cCIFAR10 (1%) with BYOL [1] and BarlowTwins [2] SSL pretraining. In both of these extra cases, uLA outperforms ERM and the SotA bias-supervised baseline. Performance with BYOL is comparable to MoCoV2+ (we have not performed extensive hyperparameter search), while performance with Barlow Twins is inferior. The latter could possibly be explained by the form of the Barlow Twins objective which attempts to match an estimation of cross-covariance between features using (spurious) training statistics to the identity.
>
> [1] Grill, Jean-Bastien, et al. "Bootstrap your own latent-a new approach to self-supervised learning." NeurIPS. 2020.
> [2] Zbontar, Jure, et al. "Barlow twins: Self-supervised learning via redundancy reduction." ICML. 2021.
>
> ### Weakness 2.a: Robustness to hyperparameters
>
> Beyond the typical hyperparameters in stochastic optimization of deep neural networks (learning rate, batch size, architecture, augmentation choices and others), we have perceived performance benefits when we search over the logit adjustment strength $\eta$, biased model calibration temperature $\tau$ and the number of epochs for training the linear head of the bias proxy network $T_\mathrm{stop}$. As we present in **Table 7 of Appendix E**, we have found stable search spaces for $\tau$ (values around 1.0, with default value 1.0) and $\eta$ (values larger than or equal to 1.0 with default value 1.0) across benchmarks. Furthermore, for $T_\mathrm{stop}$ we find in **left plot of Figure 2 in the rebuttal material** that the group-balanced test accuracy of a debiased classifier trained with uLA on cCIFAR10 (1%) increases given sufficient training time of the bias network and then it stays within the noise level of optimal performance.
>
> ### Weakness 2.b: Request for more principled method for hyperparameter search
>
> We want to highlight that one of the core contributions of our work is a meticulous methodology for model selection and hyperparameter search for robust/ood classification problems. We have demonstrated the effectiveness of our approach via experimenting against known baselines in the literature, often outperforming alternatives which use privileged group annotation information during hyperparameter search (Table 1,2,3). Extensive ablation experiments show that our hyperparameter search methodology is more reliable than a naive alternative validation objective (Figure 4).
>
> The choice of hyperparameter search algorithm can be any black-box optimizer, however in this work we opt for random search, which is the most simple, general and yet more effective than grid search [4].
>
> [4] Bergstra, James, and Yoshua Bengio. "Random search for hyper-parameter optimization." JMLR 13.2 (2012).
>
> ### Question 1: Are baselines pretrained with MoCoV2+ too?
>
> No, they do not as the methods described in their respective papers do not utilize any form of pretraining for the debiased network. We compare against accuracies reported by the peer-reviewed published methods in standard benchmarks.
>
> ### Question 2: On validating the model periodically during training and model selection
>
> If we understand the reviewer correctly, there is a concern about the practice of periodic validation of a model during its training. We want to note that this is a well-documented practice, commonly referred to as ``early stopping’’. From the Deep Learning book [8, Chapter 7]: "The only significant cost to choosing this hyperparameter automatically via early stopping is running the validation set evaluation periodically during training", referring to how to choose the number of training iterations with early stopping.
>
> While in many standardized benchmarks the number of epochs has been already searched for, in ood generalization literature it is often [9] a sensitive factor and early stopping can bring benefits in generalization. Our work addresses the need for bias-unsupervised validation for robust model selection using iid data resources. In the right plot of Figure 2 in the rebuttal material, we demonstrate that our validation accuracy [described in Equation 7] closely follows the group-balanced test accuracy and model selection according to it is likely to be close to the maximum test accuracy. In contrast, if a practitioner performs model selection by maximizing the (average) iid validation accuracy, they will get much worse performance.
>
> [8] Goodfellow, Ian, Yoshua Bengio, and Aaron Courville. Deep learning. MIT press, 2016.
> [9] Gulrajani, Ishaan, and David Lopez-Paz. "In search of lost domain generalization." ICLR. 2021.
>
> ### Question 3.a: Why did we use an ImageNet pretrained model on Waterbirds? How would SSL affect performance?
>
> For Waterbirds, we abide by the practice followed by the baselines in literature [see references at Table 2] for fair comparison. We will highlight this in the main text. All methods initialize a ResNet50 from PyTorch ImageNet-pretrained weights. A more complete explanation is that Waterbirds’ training set contains only 4795 samples, which are too few for successful supervised or SSL training from scratch. Instead, literature has chosen to test the ability of robustness algorithms in the finetuning setting. We are pleased to find that our method is not hindered by supervised pretraining in this case.

---

> > ### Comment · Reviewer_Tfdv · 2023-08-17
> >
> > I want to thank the authors for the detailed response, which addresses most of my concern. I will keep the initial positive score.

---

> > > ### Author Response · Authors · 2023-08-18
> > > **Thank you for your response**
> > >
> > > Thank you for answering to our rebuttal and for the positive assessment of our work. To further understand the weaker and the stronger points of our paper, as well as to facilitate the decision process, we would like to know which of the concerns were addressed adequately and which were not. While it is not necessary, we would appreciate if the reviewer could specify them.

---

### Official Review · Reviewer_R4tq · 2023-07-06

**Soundness:** 3 good
**Presentation:** 4 excellent
**Contribution:** 3 good
**Rating:** 6
**Confidence:** 3

**Summary:**

This work aims to evaluate and introduce a method to make classifiers perform well across subgroups of the data, focusing in particular on “spurious correlations” where one attribute correlates with the output label in the training set, but need not at test time. While most past work doing this requires “bias annotations”, or labels of the potential spurious attribute, some work aims to avoid spurious correlations without group info. This work reveals shortcomings in these methods; they introduce a new dataset (sMPI3D) where the goal is to classify the shape of an object, which is spuriously correlated with its color. They then introduce a new method, “bias-unsupervised logits”, which adjusts the output logits approximating the log probability of the label given the hidden attribute of x, without any access to the bias attribute. To approximate this quantity, the authors take representations trained with some self-supervised learning approach (they use MoCoV2+), then fine-tune these representations towards the target labels and use this as the prediction. The authors find that their method is comparable to existing methods that use bias annotations on Waterbirds, CelebA, cMNIST, and cCIFAR, and find that their method is the only one that reliably outperforms empirical risk minimization on sMPI3D.


**Strengths:**

* The paper is well written, motivated, and referenced throughout.
* The dataset sMPI3D seems like a good contribution; in particular, the authors do a good job explaining how the number of subgroups grows exponentially with the number of attributes you care about, that this means you can’t cover all groups in the training set reliably, and this dataset gives a great simple way to test how well models extrapolate to new groups (when the individual attributes are supported, but the direct product is not)
* The authors setup---no bias annotations during both training and validation---seems realistic and important for subsequent work to follow
* The results against other methods on existing benchmarks seem promising, though they’re sometimes worse than other methods.


**Weaknesses:**

* There are places where the paper could use more exposition; for example, the current paper does not devote much time to how they extract bias variable observations, or time to how they construct the dataset (the two core contributions)
* The method, while performing comparably to baselines, only offers significant improvement over previous methods on the sMPI3D, which the authors release (and thus have a lot more control over). This is somewhat ok because the task setup for sMPI3D is different (support over all individual attributes but not the direct product in training), but it would be nice to see additional experiments.


**Questions:**

* Could you provide more intuition for why training on the actual labels is a good way to estimate p(y \mid z); in particular, what properties of the pretrained representation do you need for this to work?


**Limitations:**

Addressed limitations.

---

> ### Author Rebuttal · Authors · 2023-08-10
>
> We thank the reviewer for their careful assessment of our paper. We address the raised concerns in what follows:
>
> ### Weakness 1: There are places where the paper could use more exposition: how they extract bias variable observations, or time to how they construct the dataset (the two core contributions)
>
> Thank you for pointing this out. We estimate bias attributes via a biased classifier. That is, we train a classifier with ERM without any additional approach to avoid spurious solutions that rely on biasing attributes. Since that approach is vulnerable to spurious correlations, predictions will correlate strongly with spurious features if those exist.
>
> To create the sMPI3D dataset, we use the shape of the object as the target variable, while its color is the bias. Both generative attributes can assume 6 values so that 36 combinations are possible.  A training set is created by assigning a number $C$ of colors per shape. We also make sure that all colors and shapes are represented uniformly. The test split is created by covering the shape/color combinations that **are not** observed in the training set. In our evaluation, we consider different cases with $C \in \{2, 3, 4, 5\}$.
>
> Please refer to appendices A, C and D.1 for a more in-depth exposition of both the training procedure and the construction of sMPI3D. We finally highlight that, in case of acceptance, the additional page in the camera ready version will be used for more details on the data construction along with further discussion on the effectiveness of the bias estimation approach as evidenced by Figure 1 of the rebuttal material.
>
> ### Weakness 2: Significant improvement over previous methods only on sMPI3D
>
> We would like to highlight that uLA outperforms all other methods in Table 1 in 5 out of 8 setups. Please note that in our assessment we compare against methods which may require group information during training and/or validation in order to perform well. Our method, however, outperforms them without **any access** to privileged information sources. In particular, for the cCIFAR10-1% case we outperform the state-of-the-art approach, GroupDRO (which uses group information during training and validation), with a margin of about 24% absolute accuracy.
>
> Even though these results demonstrate the effectiveness and potential of our proposal across a diverse range of scenarios, we believe our main contribution to be the improvements demonstrated in the systematic generalization case. In fact, it is fair to claim that a considerable chunk of the existing literature on robustness to spurious correlations and group fairness is expected to fail in such a case, as confirmed by our empirical assessment. We hope our contribution motivates further work in this direction given its practical relevance.
>
> ### Question 1: Could you provide more intuition for why training on the actual labels is a good way to estimate p(y \mid z); in particular, what properties of the pretrained representation do you need for this to work
>
> We simply aim for $p(y|z)$ to be such that a given class will be consistently estimated whenever a spurious feature is observed. Luckily, training a model with ERM typically satisfies that since learners tend to focus on *easy* features. Please refer to Figure 1 on the additional experiments for empirical evidence of that being the case for the cCIFAR-10 dataset.
>
> In further detail, we want to leverage the empirical observation from [1] that a biased model will predict the actual labels using the spuriously correlated (bias) variable, rather than the true feature corresponding to the class label. Using the case of cMNIST for instance, simpler predictive features (like color) are generally learnt faster than more complex predictive features (like the shape of digits). Typically training of deep neural networks will tend to use the bias attribute, when it is a simpler feature which is spuriously predictive of the target variable [2,3]. For example, it could be the case that during training most images that display the digit 5 are green. Predicting “digit 5” whenever the green feature is present in an image is sufficient to achieve minimal training errors.
>
> In **Figure 1 of the rebuttal material** we display heatmaps comparing the soft confusion matrix computed using Equation 5 versus the distribution of bias and target attributes in the training set for a cCIFAR10 (1%) experiment. We observe that the soft confusion matrix closely matches the joint distribution between the target and the bias variable.
>
> [1] Nam, Junhyun, et al. "Learning from failure: De-biasing classifier from biased classifier." NeurIPS. 2020.
> [2] Geirhos, Robert, et al. "ImageNet-trained CNNs are biased towards texture; increasing shape bias improves accuracy and robustness." ICLR. 2019.
> [3] Rahaman, Nasim, et al. "On the spectral bias of neural networks." ICML. 2019.

---

### Official Review · Reviewer_knxG · 2023-07-07

**Soundness:** 3 good
**Presentation:** 2 fair
**Contribution:** 4 excellent
**Rating:** 6
**Confidence:** 3

**Summary:**

The paper works on mitigating the impacts of spurious correlations during the risk minimization. The authors firstly introduce a systematic generalization task and illustrates existing methods implicitly made the assumption that all group combinations are represented within the training procedure. And then reveal that the importance of bias labels in model selection. Hence, the authors leverage pre-trained self-supervised models for bias information extraction, during the training and validating the debiased models.

**Strengths:**

The proposed method uLA trains a linear classifier on top of an SSL pre-trained model. The predictions are hence biased, and could be made use of model debiasing. The key advantage is that uLA does not need any group labels, neither in the training nor in the validation.

**Weaknesses:**

* (1) The presentation could be better , i.e., (a) in figure 2, it would be much better if authors could give more introduction about MPI3D. datasets; (b) Eqn (5) is a core component of uLA, it would be better if authors could explain more about the the insights of the estimate for $\hat{p}_{\text{data}}(Y, Z)$, for example, why this is a reliable estimate? Is it possible for authors to provide some empirical verification on the effectiveness of this estimation?

* (2) Personally speaking, the proposed method is somehow related to a family of distributional robustness approach, that holds the viewpoint that "fine-tune last layer" [R1] or "post-hoc adjust the model prediction" [R2] can improve the distributional robustness substantially. Such methods also rely on a well-trained model, uLA can also be categorized in this setting since uLA leverages SSL pre-trained models. Thus, it would be better to either compare or discuss the connections between uLA and this family of robust solutions.

* (3) Efficiency of uLA: SSL algorithms usually require longer training time and more computer resources, compared with classical training algorithms.

References:

[R1] Last Layer Re-Training is Sufficient for Robustness to Spurious Correlations. [ICLR'23]

[R2] Distributionally Robust Post-hoc Classifiers under Prior Shifts. [ICLR'23]

**Questions:**

* (1) The choice of ssl algorithm: I am not asking authors to run additional experiments, just curious about how uLA behaves when different SSL algorithms are utilized, if the authors already have the result.

* (2) Analysis of $\eta$: are there any analysis on the role of $\eta$ in eqn. (6), either theoretical or empirical analysis?

* (3) How is the performance of uLA on waterbirds when a SSL pre-trained model is utilized?

**Limitations:**

As mentioned in weakness 3, one limitation is **Efficiency of uLA:** SSL algorithms usually require longer training time and more computer resources, compared with classical training algorithms.

---

> ### Author Rebuttal · Authors · 2023-08-10
>
> We thank the reviewer for their detailed feedback and their questions.
>
> ### Weakness 1.a: Presentation of MPI3D
>
> We update line 73 with: “In particular, we use the ‘real’ split of MPI3D which  consists of photographs of a robotic arm that has a colored rigid object attached to its end effector. The images are captured in a way that controls their generative attributes, such as the shape of the rigid object or the position of the robotic arm.”
> In addition, we append the following at line 77: “In Appendix D.1 we describe in more detail the construction of the sMPI3D task, while in Figure 2d we illustrate an example of a systematic split.”.
>
> ### Weakness 1.b: Is Equation 5 a reliable estimate and when?
>
> We want to leverage the empirical observation from [1] that a biased model will predict the actual labels using the spuriously correlated (bias) variable, rather than the true feature corresponding to the class label. Using the case of cMNIST for instance, simpler predictive features (like color) are generally learnt faster than more complex predictive features (like the shape of digits). Typically training of deep neural networks will tend to use the bias attribute, when it is a simpler feature which is spuriously predictive of the target variable [2,3]. For example, it could be the case that during training most images that display the digit 5 are green. Predicting “digit 5” whenever the green feature is present in an image is sufficient to achieve minimal training errors.
>
> Please refer to Figure 1 of the rebuttal material where we display heatmaps comparing the soft confusion matrix computed using Equation 5 versus the distribution of bias and target attributes in the training set for cCIFAR10 (1%). We observe that the soft confusion matrix closely matches the joint distribution between the target and the bias variable.
>
> [1] Nam, Junhyun, et al. "Learning from failure: De-biasing classifier from biased classifier." NeurIPS. 2020.
> [2] Geirhos, Robert, et al. "ImageNet-trained CNNs are biased towards texture; increasing shape bias improves accuracy and robustness." ICLR. 2019.
> [3] Rahaman, Nasim, et al. "On the spectral bias of neural networks." ICML. 2019.
>
> ### Weakness 2: Related work
>
> Thanks for the pointers. Note that we already acknowledge R1 ([37]) as a bias-supervised method in L177. We update the related work with R2 as an example of bias-supervised post-hoc method. Note that R2 differs from our proposal in that it seeks to adjust logits to align a classifier's outputs with an empirical label marginal distribution given by an **annotated** validation sample. In our case, we focus on situations where such validation data are not available, and uniform accuracy over groups is expected nonetheless. More importantly, the approach in R2 would not cover cases with systematic splits such as sMPI3D since validation data, even if annotated, would still not cover all possible groups if they are indeed i.i.d. to the training set.
>
> ### Weakness 3: Efficiency of uLA
>
> We understand the concern about training an SSL model. However, we are glad to highlight that this model is re-used to initialize the de-biased network and thus, we only need to train a single model (L179). Furthermore, training a linear probe on top of extracted features or finetuning a pretrained network typically require fewer parameter updates than training from scratch.
>
> ### Question 1: How does uLA behave when different SSL pretraining methods are used?
>
> In **Table 1 of the rebuttal material**, we demonstrate results of uLA on cCIFAR10 (1%) with BYOL [3] and BarlowTwins [4] SSL pretraining. In both of these extra cases, uLA outperforms ERM and the SotA bias-supervised baseline. Performance with BYOL is comparable to MoCoV2+ (we have not performed extensive hyperparameter search), while performance with Barlow Twins is inferior. The latter could possibly be explained by the form of the Barlow Twins objective which attempts to match an estimation of cross-covariance between features using (spurious) training statistics to the identity.
>
> [3] Grill, Jean-Bastien, et al. "Bootstrap your own latent-a new approach to self-supervised learning." NeurIPS. 2020.
> [4] Zbontar, Jure, et al. "Barlow twins: Self-supervised learning via redundancy reduction." ICML. 2021.
>
> ### Question 2: Analysis of Equation 6.
>
> Proposition 2.1 (L156 and Appendix A) provides a theoretical justification of the objective in Equation 6 with $\eta = 1$. The concluding result is that by this procedure the optimal model maximizes a lower bound to the group-balanced accuracy, hence the procedure behaves in a desired way.
>
> **For why $\eta$ should be tuned**, please refer to suggestions by [5] in the last two paragraphs of Section 4.1. What they refer as $\tau$ is $\eta$ in our case. In short, calibration errors of deep neural networks [6] need to be minimal (in this case of the debiased network) and tuning this helps in mitigating them. We will clarify its role in the main paper.
>
> [5] Menon, Aditya Krishna, et al. "Long-tail learning via logit adjustment." ICLR. 2021.
> [6] Guo, Chuan, et al. "On calibration of modern neural networks." ICML. 2017.
>
> ### Question 3: Why did we use an ImageNet pretrained model on Waterbirds? How would SSL affect performance?
>
> For Waterbirds, we abide by the practice followed by the baselines in literature [see references at Table 2] for fair comparison. We will highlight this in the main text. All methods initialize a ResNet50 from PyTorch ImageNet-pretrained weights. A more complete explanation is that Waterbirds’ training set contains only 4795 samples, which are too few for successful supervised or SSL training from scratch. Instead, literature has chosen to test the ability of robustness algorithms in the finetuning setting. We are pleased to find that our method is not hindered by supervised pretraining in this case, as long as we freeze the weights of the backbone when we are training the bias proxy network.

---

> > ### Comment · Reviewer_knxG · 2023-08-21
> >
> > Thanks authors for the detailed responses, which have addressed most of my concerns. After reading the author's rebuttal and the other reviewers' comments, I would like to keep my score (6: Weak Accept).

---

### Official Review · Reviewer_Y39n · 2023-07-08

**Soundness:** 3 good
**Presentation:** 2 fair
**Contribution:** 2 fair
**Rating:** 5
**Confidence:** 4

**Summary:**

This paper introduces a new bias-unsupervised method for addressing spurious correlations, encompassing a debiasing training algorithm and a model selection paradigm. Specifically, the method employs a Self-Supervised Learning (SSL) pre-trained model as a bias proxy. The SSL model's fixed predictions are used to adjust the logit during the predictor's training phase. These predictions are also utilized in the model selection stage to adjust the weight of the sample, aiming to approximate group-balanced accuracy. Empirical evidence supports the method's efficacy, as demonstrated on several popular benchmarks, as well as a systematic generalization benchmark.

**Strengths:**

- How to design a proper bias proxy is important for the bias-unsupervised methods.
- The presentation of the method is clear and generally easy to follow.
- A variety of benchmarks are considered in the experiments and an ablation study part is included.

**Weaknesses:**

- Inconsistency. In the introduction part, one of the motivation of this work is to solve the failure of existing methods in systematic generalization. However, though empirically shown, it is not clearly introduced how the proposed method could achieve that improvement.
- Some important related works are missing.
   - There are some other existing bias-unsupervised methods, including EIIL[1], PGI[2], and SCILL[3]. Similar to the method proposed in this work, all the 3 methods utilize a frozen reference model as a bias proxy. Especially, PGI is proposed for the systematic generalization task.
   - The logit adjustment strategy is similar to the strategies in some debiasing training methods, e.g. PoE[4] and DRiFt[5].
   - A bias-unsupervised validation paradigm is also proposed in [2].

Therefore, this paper should discuss these related works and reposition its contribution.

References:

[1] Creager et al. "Environment inference for invariant learning." ICML, 2021.
[2] Ahmed et al. "Systematic generalisation with group invariant predictions." ICML, 2021.
[3] Chen et al. "When does group invariant learning survive spurious correlations?" NeurIPS, 2022.
[4] Clark et al. "Don’t take the easy way out: Ensemble based methods for avoiding known dataset biases." EMNLP, 2019.
[5] He He, Sheng Zha, and Haohan Wang. Unlearn dataset bias in natural language inference by fitting the residual. DeepLo workshop, 2019.

**Questions:**

- Why the proposed method can benefit systematic generalization, theoretically or intuitively?

**Limitations:**

Some theoretical justifications for the proposed method are lacking.

---

> ### Author Rebuttal · Authors · 2023-08-10
>
> We thank the reviewer for their feedback, recognition of the novelty of our approach and positive remarks on the presentation of our method.
>
> ### Weakness 1/Question 1:  It is not clearly introduced how the proposed method could achieve improvement in systematic generalization, intuitively or theoretically.
>
> **Theoretically**, Proposition 2.1 (L156 and Appendix A) justifies why we expect models trained with logit adjustment to systematically generalize: it optimizes for the transfer learning problem defined by Eqs 1 and 2. At the same time, systematic generalization is just an extreme case of this transfer learning problem, for which some combinations of features have 0 probability in training+validation. The scope of our analysis considers anti-causal prediction tasks for which the target variable is a generative factor. This is relevant in practice as Fig 2 suggests. We will add a sentence which clarifies the scope in Sec 2.
>
> **Intuitively** (L63), many of the existing methods will fail in a systematic generalization task, as *there are no rare samples to reweight their loss contributions or to upsample during training*. In contrast to these methods, logit adjustment avoids this pitfall. It adds the spurious $\log p(y|z)$ to the unnormalized logits of the mode, and downweights the score of the class y that is unlikely given the observed z. That is, as explained in L147-152, this additive operation will mask out the logits that are improbable to prevail under the structure of spurious correlations of the training set.
>
> ### Weakness 2: Discussion on important related works and request for repositioning our contributions.
>
> Thank you for the important references. All of them will be mentioned in Sec 5 about Related Work. In particular, [1,2,3] will also be discussed in a dedicated related work appendix section about utilizing reference models as proxies for bias information.
>
> **[1]** focus on extracting majority and minority group annotations based on a reference model, which ideally predicts based on spurious features only. In particular, they optimize for group assignments which maximally violate a relaxation of the Environment Invariance Criterion. **[3]** carefully examines various data generation processes and they propose an alternative to [1] for bias-unsupervised group design. More concretely, [3] seeks to establish conditional independence between the predictions of the bias proxy model and the target variable given the inferred groups. Our work follows more closely the approach of [6], in which a biased network is simply trained with ERM. By utilizing a frozen backbone pretrained with SSL, our approach improves on the sensitivity of [6] to the number of training steps for the bias proxy. As we demonstrate at the **left plot of Fig 2 in the rebuttal material**, the test accuracy of a uLA debiased model increases given sufficient training time of the bias proxy and after that it stays within the noise level of optimal performance.
>
> **[2]** introduce experimental settings which highlight difficulties of various training methods. They use [1] to pretrain a bias proxy and they debias with an alternative target-conditioned invariance criterion. **The authors of [2] claim that one of their colored-MNIST settings corresponds to a systematic generalization task. However we find that this is not accurate**; systematicity can only be probed for when entirely novel attribute combinations are used to generate the test samples. In their case, however, the training set contains 80% of bias-aligned samples and 20% of bias-conflicting, exposing the model to all possible combinations. In contrast, in our sMPI3D, there are always some (target, bias) combinations which are not represented in the training set at all (Fig 2d). Finally, [2] present evaluations using different model selection strategies: they use accuracy under an iid validation set, an extra ood validation set and a mixed case. However, unlike our work, they do not propose or evaluate under a specialized robust model selection methodology using bias-unsupervised iid data.
>
> ***Thus, we want to underline that to our knowledge we are the first to examine systematic generalization classifiers trained with group-robustness methods.***
>
> Furthermore, the authors of [3] perform experiments using a methodology dubbed as TEV (Training Environments Validation), which is briefly described in Appendix D.1. Iit seems related to our approach as it validates models based on inferred groups from training. The paper cites [7] as a source for TEV, however we are not able to find a methodology which matches the provided brief description in [7], making a detailed comparison and reproducibility difficult. On the contrary, we have shown that our approach often outperforms even alternatives which use group labels during model selection (Table 1,2,3). Also, an extensive search experiment shows that our selection criterion is more reliable than an alternative version of our validation score based purely on supervised learning (Fig 4).
>
> ***Overall, we believe that our work addresses explicitly an important gap in group-robustness literature [7] and proposes an effective bias-unsupervised iid robust validation protocol.***
>
> Finally, **[4]** employs a logit adjustment approach to debiasing VQA models by incorporating a bias proxy which is trained exclusively on question data. On the other hand, **[5]** exemplifies the same intuition as logit adjustment in that a second network is trained on examples that cannot be predicted already using the spurious structure. Our work instead utilizes a simple and generic logit adjustment loss, which is derived from first principles as we show in Proposition 2.1 and Appendix A.
>
> [6] Liu, Evan Z., et al. "Just train twice: Improving group robustness without training group information." ICML. 2021.
> [7] Ishaan Gulrajani and David Lopez-Paz. “In search of lost domain generalization.” ICLR. 2021.

---

> > ### Comment · Reviewer_Y39n · 2023-08-19
> > **Response to the rebuttal**
> >
> > I appreciate the authors' efforts in addressing my concerns. However, some issues remain unresolved.
> >
> > Firstly, I'd like to highlight that a test set from the colored MNIST dataset as mentioned in [2] indeed exhibits systematic shifts, as evident from Figure 1 in [2]. In the test set $T_s$, the authors "colour the test set with the biasing colours, but such that no digit is coloured with its own biasing colour". This implies entirely novel combinations. Therefore, I contest the notion that [2] examines the systematic generalization of classifiers trained with group-robustness methods.
> >
> > Secondly, while the rebuttal acknowledges the logit adjustment methods proposed in [4] and [5], the distinctiveness of the logit adjustment method introduced in this paper remains ambiguous. Specifically, how is the new method "simpler and more generic"? Does it operate under less stringent assumptions?
> >
> > I look forward to clarifications for the above issues to better appreciate the authors' contributions.

---

> > > ### Author Response · Authors · 2023-08-20
> > > **Thank you for the response**
> > >
> > > We thank the reviewer for participating in the discussion phase. We proceed to address the remaining issues:
> > >
> > > ### Systematic Generalization in [2]
> > >
> > > Considering that we agree on that a systematic generalization benchmark needs to exhibit entirely novel combinations of attributes in the test set samples, we disagree that [2] examines such a case. This is already evident in Fig. 1. Specifically, Fig 1.a presents examples from the training set. Last row contains samples of digit 9, while it suggests that the biasing/dominant color for 9 is fuschia. Notice how a cyan 9 (which is not the dominant color) also appears in the training set. At the same time Fig 9.c presents samples from the systematic generalization test set. Observe that it also contains a cyan 9 in the last row.
> > >
> > > This is not a coincidence. The cited line from the paper in the comment just means that the test set contains only bias-conflicting samples; that is samples which are generated by combinations of (digit, not biasing color of that digit), like non-fuschia 9s. It does not claim, however, that such combinations are not exposed in the training set. In fact, "own biasing color" can just mean "highly correlated with the digit in the training set", as it is the case in [2] and for example in [8]. To underline this, we repeat here the training set generation procedure as it is presented in [2], and also mentioned in our rebuttal:
> > >
> > > > COLOURED MNIST: Consider an illustrative dataset with coloured MNIST digits. For the training set, Tr, MNIST digits are coloured with a set of digit-correlated “biasing” colours 80% of the time, and with ten random colours that are different from the biasing colours the remaining 20% of the time.
> > >
> > > We hope that this makes clear that **the training set of [2] is exposed to all possible combinations of digits and colors**, and hence under their training set construction there can be no possible test set containing samples from entirely novel combinations. Thus, in contrast to our work (see Fig 2.d and Appendix D.1 of our work), [2] does not actually examine a systematic generalization task, but an easier group-robustness case as it is further elaborated in our rebuttal.
> > >
> > > [8] Nam, Junhyun, et al. "Learning from failure: De-biasing classifier from biased classifier." NeurIPS. 2020.
> > >
> > > ### In relation to [4] and [5]
> > >
> > > [4] operates under the additional assumption that input data is available in pairs of (true context, bias source). In their case (VQA tasks) true context refers to an image and the biasing source is the textual question. In our work, true predictive features and bias features are not already "disentangled" like that in an input observation $x$. In that, our method is *simpler and more generic*.
> > >
> > > On the other hand, [5] does not have a mechanism which prevents the biased model from fitting exactly the biased training set. If the biased networks fits entirely the training set, consequent logit adjustment will not be effective as it no longer a proxy for $p(y|z)$, but on the training samples it is simply acts as look-up table for the training set. To prevent that, [5] depends on early-stopping hyperparameters for the biased network, which are tuned using unspecified data resources, possibly a balanced validation set. In contrast, our work tunes hyperparameters based on our proposed robust model selection criterion using iid data, as well as our pretraining approach based on SSL and linear probing yields decreased sensitivity to the number of training steps of the biased network (Left plot Fig. 2 of rebuttal material).

---

> > > > ### Comment · Reviewer_Y39n · 2023-08-22
> > > > **Response to the rebuttal**
> > > >
> > > > I would like to thank the authors for the response.
> > > >
> > > > Regarding the first point, I agree with the authors that the colored MNIST in [2] does not present wholly unique combinations of attributes in its test set samples. The crux, however, is to ascertain if the methodology from [2] can be adapted for systematic generalization scenarios. If the approach from [2] falls short in this aspect, whereas the new method presented in this paper excels, it would convince the novelty of the proposed method and its significance for systematic generalization. This distinction, however, remains ambiguous in the current discussion.
> > > >
> > > > For the second point, I understand that the design of the bias proxy in this paper, coupled with the model selection approach, undeniably simplifies the process and offers a more generalized solution. My primary concern pertains to the novelty of the logit adjustment step, specifically the one induced by equation (4) in this paper. It's essential to elucidate the theoretical disparities between this equation and those proposed in other works.
> > > >
> > > > In summation, I believe the paper can be further refined. I would like to encourage the authors to address these concerns to amplify the potential impact and utility of their work for the broader community.

---

> > > > > ### Author Response · Authors · 2023-08-22
> > > > > **Discussing further**
> > > > >
> > > > > We would like to thank the reviewer for maintaining engagement with us during the discussion phase.
> > > > >
> > > > > ### First point
> > > > >
> > > > > ***We are glad to find that the reviewer agrees with us on the original contested point: [2] does not examine a systematic generalization task***, in which the test set consists of samples generated from entirely novel combinations of attributes.
> > > > >
> > > > > We would have been happy to provide with the extra quantitative experiments, if that was the original request of the reviewer at the beginning of rebuttal period when we could have the time to respond. Nonetheless, current evidence in our paper suggests that methods which depend on reweighting per-sample losses, like the one presented at [2], cannot deal with systematic generalization tasks. The reason is that rare examples corresponding to the novel combinations cannot be found during training, thus these methods will not find useful sanples to reweight. For example, as we present in Table 3, reweighting methods such as GroupDRO and Just-train-Twice perform subpar in the sMPI3D task  compared to our method, and in particular they perform close to the percentage of combinations found in the training set (out of the total possible ones).
> > > > >
> > > > > ### Second point
> > > > >
> > > > > ***We would like to thank the reviewer for highlighting that our method "undeniably simplifies the process and offers a more generalized solution."*** Furthemore, we would like to remind that, while [4] and [5] do not offer with any theoretical justification, our work is deriving the general form of logit adjustment for the spurious correlation problem from first principles, as we have already argued in the original rebuttal response (Proposition 2.1 and Appendix A). Finally, we would like to remind that the logit adjustment paradigm itself originates from the long-tail data learning literature [9], as we present in the first sentence of the Logit Adjustment paragraph in Sec. 2 , and our work effectively adapts it to the spurious correlation problem while providing with theoretical justifications.
> > > > >
> > > > > **In any case, our work does not claim novelty of the logit adjustment technique. What our work proposes is a training+validation methodology for group-robust classifiers without any group information**. For that we offer a favorable alternative (which uses logit adjustment in a well-justified manner) to methods which implicit rely on meta-information about existence of all groups during training (systematic gen. tasks) or explicitly to ood resources for successful model selection.
> > > > >
> > > > > [9] Menon, Aditya Krishna, et al. "Long-tail learning via logit adjustment." arXiv preprint arXiv:2007.07314 (2020).
> > > > >
> > > > > ### Final note
> > > > >
> > > > > During the rebuttal period, we believe that we have sufficiently addressed all points raised by the reviewer, as they have also reached agreement on those points. We will update the paper to reflect and clarify the questions raised by the reviewer and we will also discuss the extra related work, as we have meticulously done in our rebuttal.

---

### Author Rebuttal · Authors · 2023-08-10

We thank all reviewers for their insightful assessment of our work and for providing useful feedback and actionable suggestions. We are glad they found the method we proposed to be simple yet effective (reviewer Tfdv) and efficient since no bias labels are required during training or validation (reviewer knxG), our evaluation broad covering a number of benchmarks and including ablations (reviewer Y39n) and realistic (reviewer R4tq), the sMPI3D data to be a good contribution (reviewer R4tq), and the manuscript to be clear (reviewers Y39n and R4tq).

We addressed each reviewer individually and provided additional empirical analysis as suggested. New results include:
  -  Evaluations of models using SSL backbones pre-trained with different approaches.
  -  Analysis of the effectiveness of using ERM trained classifiers as estimators of bias attributes.
  -  A comparison of different cross-validation criteria show how i.i.d. accuracy yields to sub-optimality and that our proposal closely aligns with the oracle group-balanced test accuracy, as suggested by our original results.
  -  Plot on how bias proxy training time influences group-balanced test accuracy of a debiased model.

Furthermore, we summarize the edits that we suggest to perform in the paper as a result of reviewer feedback:
  -  Clarification on the scope and relevance to systematic generalization for Proposition 2.1
  -  Updated related work and extra Appendix section which discusses methods to create bias proxy networks
  -  More main paper details on MPI3D and sMPI3D
  -  Extra study in main paper: evidence about Eqn 5 being reliable (Fig 1 of rebuttal material)
  -  Clarification that ImageNet pre-training for Waterbirds is for fair comparison to baselines
  -  Clarification on why we search $\eta$
  -  Add in Appendix the ablation for different SSL pretraining methods (Table 1 of rebuttal material)
  -  Add in Appendix the sensitivity curve: bias proxy training time vs test accuracy of debiased model (Left plot of Fig 2 in rebuttal material)
  -  Add in Appendix the validation and test curves during training (Right plot of Fig 2 in rebuttal material)

---

### Author Response · Authors · 2023-08-15
**Request to participate in discussion phase**

Dear reviewers and AC,

We appreciate the reviewers for dedicating time and effort to thoroughly evaluate our work. We have taken care to meticulously address all of their concerns and questions, providing detailed comparisons to related work and extended empirical evidence reinforcing the claims of our work. As the discussion phase is about to end and we are still anticipating for feedback corresponding to it, we would like to kindly invite the reviewers to participate in this phase and engage in our responses. In the case of any follow-up questions, we will in turn provide with further clarifications.

Kindly,
The authors

---

### Decision · Program_Chairs · 2023-09-21

**Decision:**

Accept (poster)

**Comment:**

Models trained with ERM can underperform on under-represented subgroups due to the presence of spurious correlations or biases. Existing approaches for tackling this problem assume the presence of all label-bias combinations to be present in either the training or validation set. In contrast, this paper seeks to learn a classifier robust to spurious correlations in an entirely bias-unsupervised manner. They do so by training a proxy model to predict the "bias" attribute and using predictions from this model to formulate a logit-adjusted training loss.

The reviewers were generally appreciative of the problem setup and recognize the introduction of the MPI3D as a useful contribution.
While there were some initial concerns about the exposition and lack of clarity about how the paper trains a reliable proxy model, these were resolved during the rebuttal.

However, an important pending concern was about the **paper's claim that it guarantees "systematic generalization"** (Reviewer Y39n), i.e. the ability to generalize to combinations of attribute not seen in the training/validation sample. The authors point to the use of "logit adjustment" to guarantees this (they say that in contrast, prior work are unable to systematically generalize as they instead rely on *re-weighting* rare samples).

It took me quite a while to convince myself of the authors claim (after going through proofs/material in the appendix). While we are happy to accept this paper, we urge the authors to provide a stronger justification for their claim. Please take a close look at the following comments.

**Closer inspection at the claim of systematic generalization:**

At the heart of the author's claim is the assumption in eq. 14, which essentially states that the conditional distribution $p_{\rm data}(y | x, z)$ can be decomposed as:

$$
p_{\rm data}(y | x, z) = p_{\rm data}(y | z) \cdot G(x, y),
$$

for some function $G$ independent of $z$. In the paper, $G(x, y) \propto \exp(h_\theta(x)_y)$. In other words, the authors assume that $\frac{ p\_{\rm data}(x | z, y)  }{ p\_{\rm data}(x | z) } = \frac{ p\_{\rm data}(y | x, z)  }{ p\_{\rm data}(y | z) }$ is independent of $z$. Unfortunately, this assumption was not made very explicit, except in passing in Proposition 2.1. Please state this explicitly in the main text.

I can see how under this assumption, the use of logit-adjustment guarantees systematic generalization. Specifically, due to the independence assumed, we don't need to see all $(y, z)$ combinations to learn $G(x, y)$ (or a function $h_\theta(x)$ that shares the same maximizer over $y$).

However, I am still unconvinced why re-weighting the loss woudn't have the same effect (assuming eq. 14 holds). More precisely, wouldn't re-weighting by the inverse of $p_{\rm data}(y | z)$ instead of using it as a part of the logit-adjustment also be able to learn $G(x,y)$? For example, if we were to use the following re-weighted loss:
$$
\\mathbb{E}\_{x, y}\\left[ \\frac{1}{\\hat{p}\_{\\rm data}(y | z)} \\cdot \\ell_{\\rm ce}(y, h\_{\\theta}(x)) \\right],
$$
even though the loss will not see some $(y, z)$ combinations, wouldn't it still end up learning $h\_{\\theta}(x) \approx \log\left( \frac{ p\_{\rm data}(y | x, z)  }{ p\_{\rm data}(y | z) } \right)$, given the assumption that the RHS is independent of $z$.

The authors are strongly encouraged to make their assumptions clear and explain why they aid in systematic generalization. If the authors do believe that prior re-weighting style approaches would not guarantee systematic generalization under the same assumptions, they need to explain this clearly in the paper.